# VIDEO MODELS ARE ZERO-SHOT LEARNERS AND REASONERS

## ABSTRACT

The remarkable zero-shot capabilities of Large Language Models (LLMs) have propelled natural language processing from task-specific models to unified, generalist foundation models. This transformation emerged from simple primitives: large, generative models trained on web-scale data. Curiously, the same primitives apply to today's generative video models. Could video models be on a trajectory towards general-purpose *vision* understanding, much like LLMs developed general-purpose *language* understanding? We demonstrate that Veo 3 can solve a broad variety of tasks it wasn't explicitly trained for: segmenting objects, detecting edges, editing images, understanding physical properties, recognizing object affordances, simulating tool use, and more. These abilities to perceive, model, and manipulate the visual world enable early forms of visual reasoning like maze and symmetry solving. Veo's emergent zero-shot capabilities indicate that video models are on a path to becoming unified, generalist vision foundation models.

Project page: https://zero-shot-anonymous.github.io

## 1 INTRODUCTION

We believe that video models will become unifying, general-purpose foundation models for machine vision just like large language models (LLMs) have become foundation models for natural language processing (NLP). Within the last few years, NLP underwent a radical transformation: from task-specific, bespoke models (e.g., one model for translation, another one for question-answering, yet another one for summarization) to LLMs as unified foundation models. Today's LLMs are capable of general-purpose language understanding, which enables a single model to tackle a wide variety of tasks including coding [1], math [2], creative writing [3], summarization, translation [4], and deep research [5, 6]. These abilities started to emerge from simple primitives: training large, generative models on web-scale datasets [e.g 7, 8]. As a result, LLMs are increasingly able to solve novel tasks through few-shot in-context learning [7, 9] and zero-shot learning [10]. Zero-shot learning here means that prompting a model with a task instruction replaces the need for fine-tuning or adding task-specific inference heads.

Machine vision today in many ways resembles the state of NLP a few years ago: There are excellent task-specific models like "Segment Anything" [11, 12] for segmentation or YOLO variants for object detection [13, 14]. While attempts to unify some vision tasks exist [15–26], no existing model can solve *any* problem just by prompting. However, the exact same primitives that enabled zero-shot learning in NLP also apply to today's generative video models—large-scale training with a generative objective (text/video continuation) on web-scale data [27]. In this article, we therefore ask: Do video models develop general-purpose *vision* understanding, similar to how LLMs developed general-purpose *language* understanding? We answer this question in the affirmative:

1. Analyzing 18,384 generated videos across 62 qualitative and 7 quantitative tasks, we report that Veo 3 can solve a wide range of tasks that it was neither trained nor adapted for.

2. Based on its ability to **perceive**, **model**, and **manipulate** the visual world, Veo 3 shows early forms of "chain-of-frames (CoF)" **visual reasoning** like maze and symmetry solving.

3. While task-specific bespoke models still outperform a zero-shot video model, we observe a substantial and consistent performance improvement from Veo 2 to Veo 3, indicating a rapid advancement in the capabilities of video models.

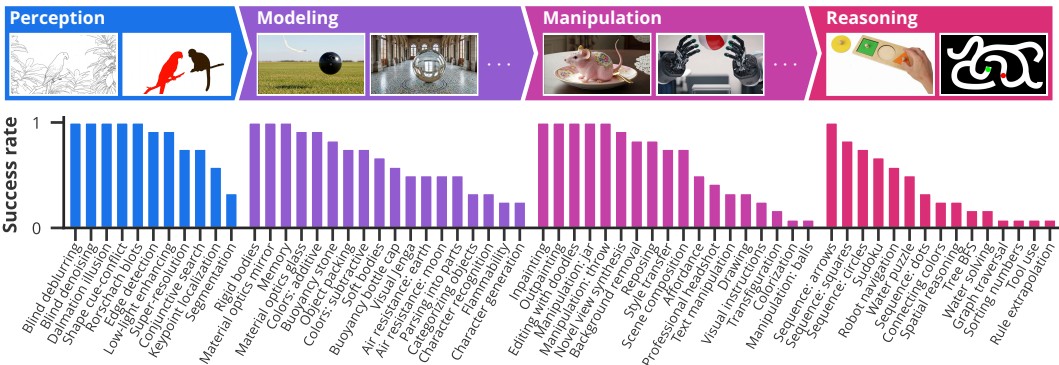

Figure 1: **A qualitative overview of Veo 3's zero-shot abilities**. The plot shows Veo 3's success rate across 12 samples as a rough estimate of model performance on 62 tasks across the vision stack. Tasks are described in Sec. 3 and shown in App. A. Find videos of all tasks on our project page.

## 2 METHODS

**Approach and motivation**   Our method is simple: We prompt Veo. This minimalist strategy is intentional, as it mirrors the transformation of NLP from task-specific fine-tuning or training to prompting a capable foundation model [28–30]. Here, we adopt the same philosophy to explore the capabilities of Veo 3 as a general-purpose vision model. Following Bommasani et al. [31, p. 3], a foundation model is any model "trained on broad data (generally using self-supervision at scale)" which can be adapted (here: via prompting) "to a wide range of downstream tasks".   NEW NEW

> **Takeaway 1**   In NLP, prompting replaced task-specific training or adaptation for many tasks. A similar paradigm shift is on the horizon in machine vision, facilitated by video models.

**Video generation**   For each task, we query the publicly available Veo 2 or Veo 3 models via Google Cloud's Vertex AI API. We prompt the model with an initial input image (which the model uses as the first frame) and a text instruction. The models then generate a 16:9 video at 720p resolution, 24 FPS, for a duration of 8s. Veo 3 has model ID `veo-3.0-generate-preview` and Veo 2 model ID `veo-2.0-generate-001`. According to the Vertex documentation [32], the API uses an LLM-based prompt rewriter. This means that for some tasks, the solution is likely to come from the LLM instead of the video (e.g., Fig. 55: Sudoku). We treat the system (rewriter and video generator) as a single black-box entity. However, to isolate the video model's reasoning abilities, we verified that a standalone LLM (Gemini 2.5 Pro [2]) could not reliably solve key tasks (Fig. 58: Robot navigation, Sec. 4.5: Maze solving, Sec. 4.6: Visual symmetry) from the input image alone.

**Why Veo?**   The core argument of this paper—that video models are zero-shot learners and reasoners—can be supported by demonstrating success on *any* sufficiently capable model.   We choose Veo for its consistent high ranking on `text2video` and `image2video` leaderboards [33]. Unless noted otherwise, figures are generated with Veo 3. To provide a sense of how rapidly performance is improving, our quantitative analyses compare Veo 3 with its predecessor, Veo 2, released roughly half a year earlier: Veo 2 was announced in December 2024 and released in April 2025 [34, 35], while Veo 3 was announced in May 2025 and released in July 2025 [36, 37].

## 3 QUALITATIVE RESULTS: SPARKS OF VISUAL INTELLIGENCE?

We begin with a comprehensive, qualitative investigation across visual tasks to assess the potential of video models as visual foundation models. We organize our findings into four hierarchical capabilities, each building on the preceding ones (c.f. Fig. 1 and Fig. 2):

    1. **Perception** as a foundational ability to understand visual information.

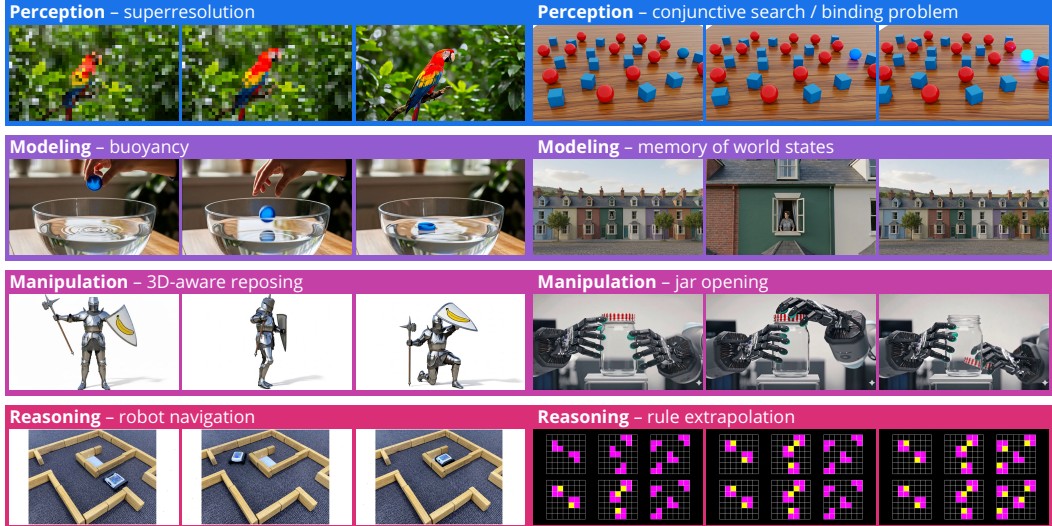

Figure 2: **Veo 3 zero-shot learning and reasoning examples**. From classic **perceptual** tasks (superresolution, visual search) to **modeling** (buoyancy, memory of world states after zooming in), **manipulation** (pose editing, simulating dexterous manipulation) and **visual reasoning** (navigation, rule extrapolation): Veo 3 can zero-shot solve a host of visual tasks that are specified as an input image and a text prompt. Examples are shown in App. A; videos of all tasks are on our project page.

2. **Modeling**, which builds on the perception of objects to form a model of the visual world.

3. **Manipulation**, which meaningfully alters the perceived and modeled world.

4. **Reasoning** across space and time over a sequence of manipulation steps.

While capability boundaries often overlap, this hierarchy provides a framework for understanding the emergent abilities of video models. For example, solving a maze (see Fig. 57 and Sec. 4.5) requires perceiving the maze, modeling its state (walls vs. floor), and finally manipulating an object (a mouse, a circle) to move from start to finish.

For each task in this section, we prompt Veo 3 22 times and report the *success rate* in the caption. FIX *Success rate* is the fraction of videos that solve a task, as determined by human annotators (see App. A.5). A success rate greater than 0 suggests that the model possesses the *ability* to solve the task, while a success rate closer to 1 indicates that the task is solved *reliably* irrespective of the random seed. While not a substitute for the systematic quantification we perform in Sec. 4, this provides a ballpark estimate of the model's capabilities.

**Perception** Computer vision has historically relied on a suite of specialized models for tasks like segmentation [11, 12], object detection [13, 14], and edge detection [38]. While some backbones can be adapted or fine-tuned for other tasks, training-free transfer to novel tasks is rare, limiting generalization. As we show here, this is changing with large video models.

Without any task-specific training, Veo 3 can perform a range of classic computer vision tasks, including edge detection (Fig. 10), segmentation (Fig. 11), keypoint localization (Fig. 12), superresolution (Fig. 13), blind deblurring (Fig. 14), denoising (Fig. 15) and low-light enhancing (Fig. 16). Some of these tasks were already demonstrated in [39], but Veo's emergent abilities extend farther: It can perform complex tasks like conjunctive search (Fig. 17) and interpreting ambiguous images such as the classic dalmatian illusion (Fig. 18), the cat shape in a texture-shape cue conflict image (Fig. 19), and colored blots from the Rorschach test (Fig. 20). Apart from denoising—the classic diffusion objective—none of these tasks are explicitly trained for in video models.

> **Takeaway 2** Veo 3 shows emergent zero-shot perceptual abilities well beyond the training task. Just like LLMs replaced many task-specific NLP models, video models will likely replace most bespoke models in computer vision—once they become sufficiently cheap and reliable.

**Modeling: intuitive physics & world models** Based on their *perception* of the visual world, video models are starting to *model* it, too. Modeling the world and the principles that govern it (e.g., laws of physics) is a critical step toward successful prediction and action. Several works have investigated and quantified intuitive physics in deep models [e.g., 40–54]. Here, we investigate an exemplary subset of tasks from these works. Veo's grasp of physics is demonstrated by its ability to model various physical properties, like flammability (Fig. 21), rigid and soft body dynamics and their surface interactions (Fig. 22) and air resistance affecting falling objects (Fig. 23), and buoyancy (Fig. 24). As illustrated by the Visual Jenga task [55], Veo can remove objects from a scene in a physically plausible order (Fig. 25) and understands which objects fit into a backpack (Fig. 26). Furthermore, it correctly generates some optical phenomenona like refraction and reflection (Fig. 27) and additive/subtractive color mixing (Fig. 28). Beyond physical characteristics, Veo understands some abstract relationships which is an important aspect of modeling the world. For example, Veo can distinguish categories like toys from other objects like a laptop (Fig. 29). On samples inspired by the Omniglot dataset [56], we demonstrate Veo's ability to recognize patterns, generate variations thereof, and parse larger wholes into parts (Fig. 30). Lastly, Veo maintains a memory of the world state across time and camera movements within the video context (Fig. 31).

**Manipulation: editing & imagination** Based on its ability to *perceive* objects and *model* their relation to each other and the world, Veo can meaningfully *manipulate* the visual world, too. This enables Veo 3 to perform a variety of zero-shot image editing tasks like background removal (Fig. 32), style transfer (Fig. 33), colorization (Fig. 34), inpainting (Fig. 35), and outpainting (Fig. 36). Furthermore, it can manipulate text elements (Fig. 37), and edit images based on doodle instructions (Fig. 38). Veo's understanding of 3D world enables it to compose scenes from individual components (Fig. 39), generate novel views of objects and characters (Figs. 40 and 41), smoothly transform one object into another (Fig. 42), or change the perspective, lighting, and appearance to turn a selfie into a professional photograph (Fig. 43).

This ability to plausibly modify a scene allows it to imagine complex interactions, simulate dexterous object manipulation (Fig. 44; note that we do not test actual robots as e.g. [57] do), interpreting object affordances (Fig. 45), demonstrating how to draw a shape (Fig. 46) and roll a burrito (Fig. 47). Overall, video models can meaningfully manipulate and simulate aspects of the (digital) visual world.

**Visual reasoning across time and space** *Perception*, *modeling*, and *manipulation* all integrate to tackle *visual reasoning*. While language models manipulate human-invented symbols, video models can apply changes across the dimensions of the real world: time and space. Since these changes are applied frame-by-frame in a generated video, this parallels chain-of-thought in LLMs and could therefore be called *chain-of-frames*, or CoF for short. In the language domain, chain-of-thought enabled models to tackle reasoning problems [28], and visualizing intermediate steps helps [58]. Similarly, chain-of-frames (a.k.a. video generation) might enable video models to solve challenging visual problems that require step-by-step reasoning across time and space.

We see early signs of this ability in tasks such as generating a valid graph traversal (Fig. 48), performing visual breadth-first search on a tree (Fig. 49), completing visual sequences (Fig. 50), connecting matching colors (Fig. 51), fitting shapes into holes (Fig. 52), and sorting numbers (Fig. 53). Furthermore, Veo can use tools to accomplish a visual task (Fig. 54) and solve simple Sudokus (Fig. 55) or visual puzzles (Fig. 56). Finally, it can solve mazes and navigation tasks (Figs. 57 and 58 and Sec. 4.5) and extrapolate rules from visual examples (Fig. 59). While not always perfect, the model's ability to solve such problems in a zero-shot manner points to exciting possibilities for more advanced visual reasoning and planning in future, with more capable video models.

> **Takeaway 3** Frame-by-frame video generation parallels chain-of-thought in language models. Just like chain-of-thought (CoT) enables language models to reason with symbols, a "chain-of-frames" (CoF) enables video models to reason across time and space.

**Summary** Taken together, the qualitative examples from this section indicate that a capable video model like Veo 3 possesses strong zero-shot learning abilities. While the results are not always perfect, the model consistently demonstrates the capacity to solve a wide variety of tasks for which it was not explicitly trained.

## 4 QUANTITATIVE RESULTS

The previous section offered a qualitative exploration of video model capabilities. In this section, we add a quantitative assessment for seven tasks. As in Sec. 3, we consider different facets of visual understanding: For **perception**, we assess Veo on edge detection and segmentation. For **manipulation**, we examine image editing and object extraction performance. Finally, we evaluate **reasoning** abilities through maze solving, visual symmetry, and visual analogies. We do not include evaluations for **modeling**, since this area is well addressed by recent benchmarks, see Sec. 3.

We evaluate performance separately for the *best frame* and the *last frame* (where applicable). *Best frame* reports the best performance across any frame in the generated videos. This indicates the performance ceiling, but the optimal frame is not known a priori. Therefore, we also report performance on the *last frame* of each video, which may underestimate a model's ability but has the practical advantage that the frame is predetermined. This distinction is important because Veo tends to continue animating a scene even after task completion, potentially reducing last frame performance.

Where applicable, we use the state-of-the-art image editing model Nano Banana [59] as a reference. As a general trend, we observe a large performance increase from Veo 2 to Veo 3, often matching or exceeding Nano Banana's performance. Performance tends to improve substantially from $k = 1$ to $k = 10$ attempts, indicating that a good solution can be found in a reasonable number of tries. While image models are excellent zero-shot learners, too [60–63], video models are the more general framework because of their ability to process both temporal and spatial information.

### 4.1 PERCEPTION: EDGE DETECTION

Despite not being trained for it, Veo 3 can be prompted to detect, and therefore **perceive**, edges. Fig. 3 details edge detection performance (measured by OIS; details and prompt in App. B.1) for Veo 2 and Veo 3. While Veo 3 (0.77 pass@10) is not on par with task-specific SOTA (0.90) [65], its performance is remarkable for two reasons: First, it is zero-shot. Second, many of Veo 3's edge maps are more detailed than the ground truth. For example, Veo 3 correctly outlines foliage and tire profiles; this hurts performance on the dataset but seems more indicative of a dataset limitation than a model limitation (the annotators understandably did not bother to trace each and every edge).

### 4.2 PERCEPTION: SEGMENTATION

Instance segmentation requires delineating (i.e., **perceiving**) distinct objects in an image. Contrary to classic instance segmentation or promptable segmentation, we prompt models to segment all objects in a scene, without specifying an object category or location. We report mean Intersection over Union (mIoU); experiment details are in App. B.2. As shown in Fig. 4, Veo 3 achieves an mIoU of 0.74 (best frame pass@10), comparable to Nano Banana's 0.73. Naturally, Veo 3 lacks behind the performance of a bespoke model like SAMv2 [12], but nevertheless shows remarkable zero-shot segmentation abilities. Interestingly, the prompt really matters: Veo consistently performs better with a green background than a white one (0.74 vs. 0.66 best frame pass@10); possibly due to the widespread use of green screens. See also App. C for prompting best practices.

### 4.3 MANIPULATION: OBJECT EXTRACTION

Can Veo perceive and extract (i.e., **manipulate**) all objects in a scene? We test this using a simple dataset depicting one to nine animals (details in App. B.3). Veo is asked to extract and line up all

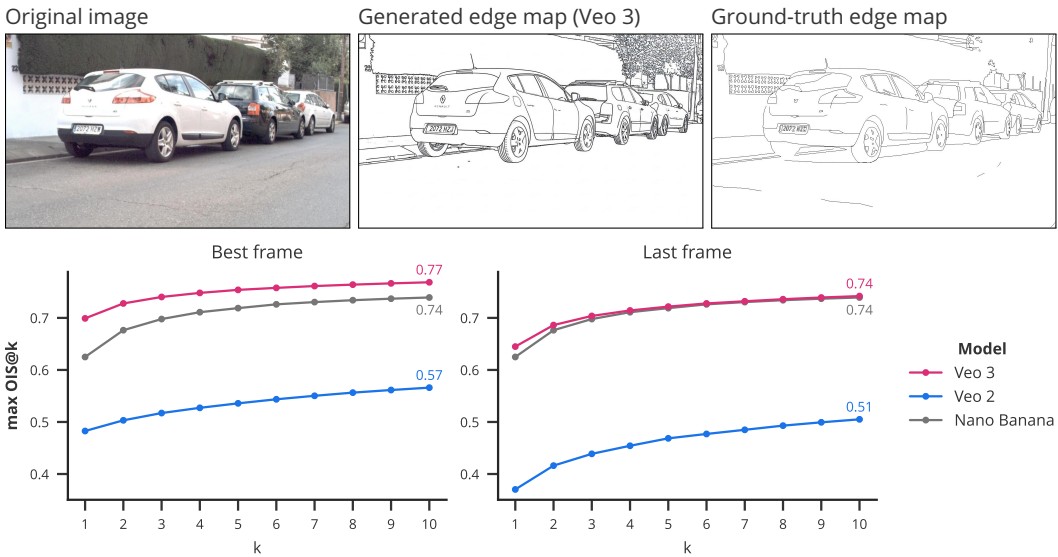

Figure 3: **Edge detection** on all 50 test images from BIPEDv2 [64, 65]. We generate 10 videos per sample and report best performance over $k$ attempts as a function of $k$. Prompt: *"All edges in this image become more salient by transforming into black outlines. Then, all objects fade away [...]"* Details & full prompt: App. B.1.

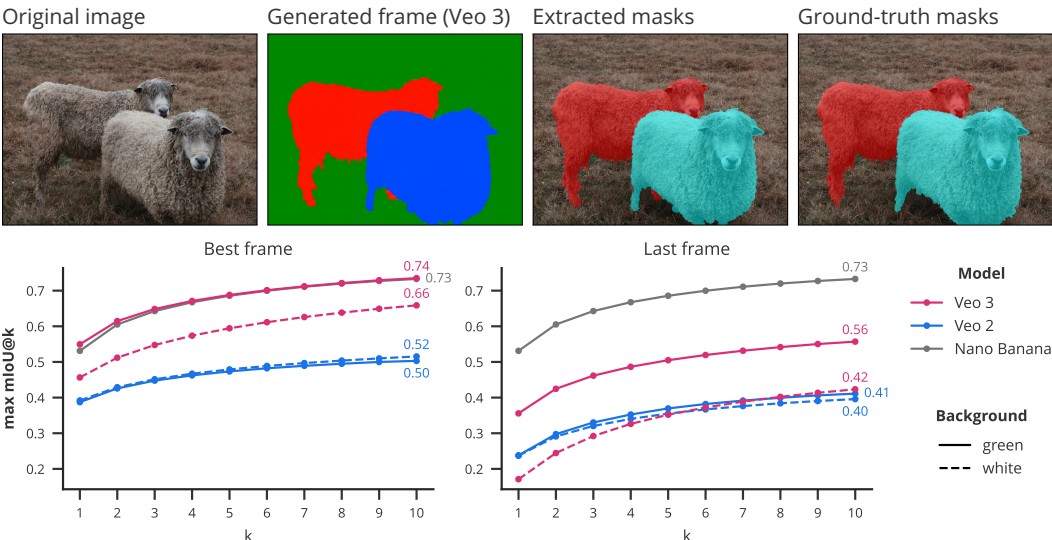

Figure 4: **Class-agnostic instance segmentation** on a subset of 50 easy images (1-3 large objects) from LVIS [66]. Prompt: *"[...] each distinct entity is overlaid in a different flat color [...] the background fades to {white, green} [...]"* Details & full prompt: App. B.2.

animals horizontally, with white space between them (in some sense, a visual "tally"). To assess whether the number of extracted animals is correct, we count connected components in the last frame. Fig. 5 shows an example. While Veo 2 performs around chance, Veo 3 achieves up to 93% pass@10. Given the simplicity of the task, a perfect model should easily achieve 100% accuracy.

## 4.4 MANIPULATION: IMAGE EDITING

Image editing requires **manipulating** images according to a text instruction (e.g., adding/removing objects or changing their appearance). We evaluate whether Veo can edit images on a random subset

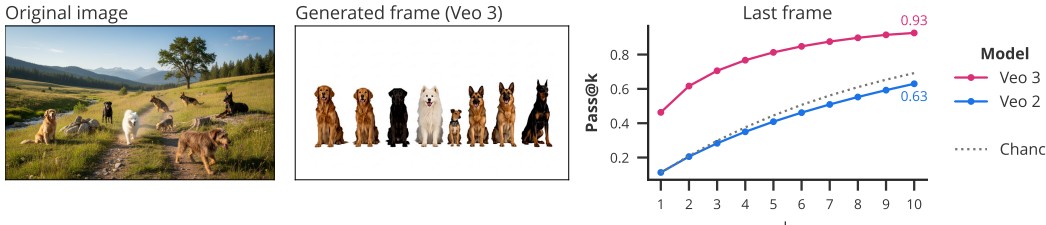

Figure 5: **Object extraction** on an animal dataset. Prompt: *"The background changes to white [...] all animals line up in a row [...]"* Details & full prompt: App. B.3.

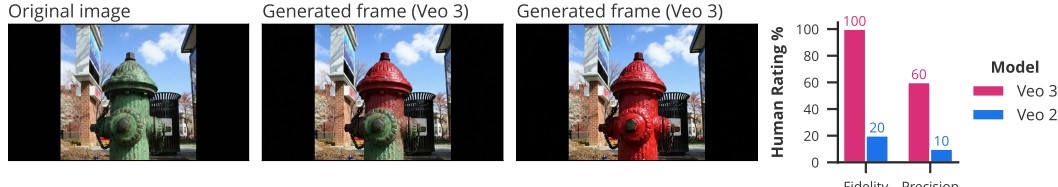

Figure 6: **Image editing** on a subset of Emu-edit [67]. Prompt: *"Create a smooth, static animation that slowly changes the color of the fire hydrant to red. [...]"* Details & full prompt: App. B.4.

of 30 samples from Emu-edit [67]. Veo has a strong bias for animated scenes and might introduce unintended changes (e.g., camera movement, animating people). We therefore ask three human raters to evaluate *fidelity* (correct edit) and *precision* (correct edit without unintended changes). An example edit and results are shown in Fig. 6. We find that Veo 3 especially excels in preserving details and textures across edits. If unintended changes such as camera movement or animating people can be controlled better, video models could become highly capable 3D-aware image and video editors (see also [24, 39, 68, 69]).

## 4.5 REASONING: MAZE SOLVING

Maze solving tests a model's ability to plan a path in a constrained environment, a key aspect of **reasoning**. In our setup, a red circle needs to navigate to a goal (green circle) without crossing any walls. We automatically verify the path (details in App. B.5) and present results for different mazes in Fig. 7. Veo 3 shows zero-shot maze solving abilities, significantly outperforming Veo 2 which often produces illegal moves. For instance, in 5×5 grids, Veo 3 achieves a pass@10 rate of 78% compared to Veo 2's 14%. The consistent performance gap highlights the advancing reasoning capabilities of Veo 3. While Nano Banana matches or surpasses Veo 3's performance on rectangular mazes, it fails to solve irregular mazes entirely. Similarly, Gemini 2.5 Pro outperforms Veo 3 on small mazes when given an ASCII representation of the maze (T2T), but falls behind on 9×9 mazes, and generally struggles when the maze is represented as image (as opposed to text) input. Both comparisons highlight the advantages of solving a visual task step-by-step in a visual medium.

## 4.6 REASONING: VISUAL SYMMETRY SOLVING

Completing a pattern to be symmetrical assesses the ability to understand and apply spatial **reasoning**. We create a custom dataset of shapes (e.g., heart, letters) and random patterns (see App. B.6 for details). Fig. 8 shows that Veo 3 outperforms Veo 2 and Nano Banana by a large margin. We also use this task to systematically analyze how different prompts affect performance in App. C: The pass@1 difference between best and worst prompt is 40 percentage points on the shape split and 64 percentage points on the random split.

## 4.7 REASONING: VISUAL ANALOGY COMPLETION

Visual analogies test a model's ability to understand transformations and relationships between objects, a form of abstract **reasoning**. Concretely, we prompt the model to fill the missing quadrant

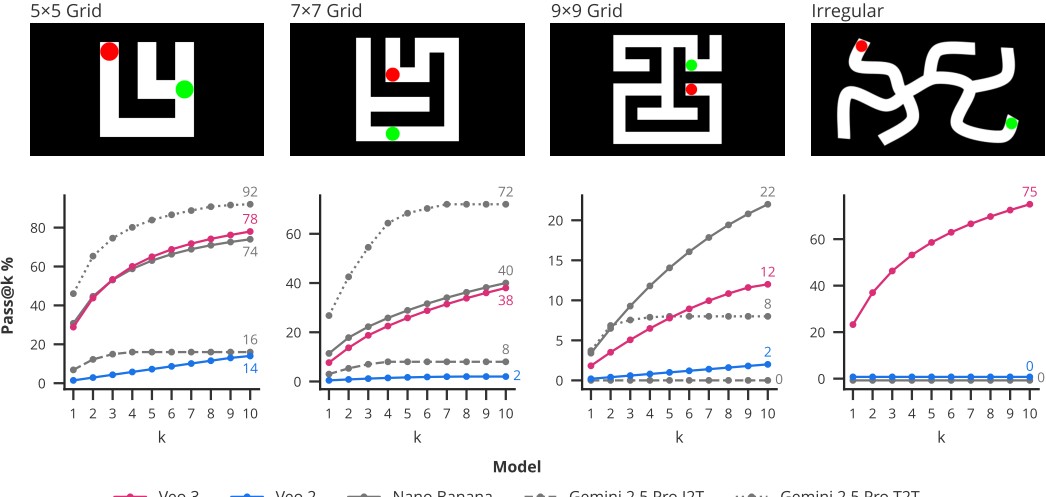

Figure 7: **Maze solving**. Mazes of various sizes with start (red) and goal (green) locations. Prompt: *"[...] The red circle slides smoothly along the white path, stopping perfectly on the green circle [...]"* Details & full prompt: App. B.5. Veo 2 struggles to solve even small sizes, mostly due to illegal moves early in the generation. Veo 3 performs much better and benefits from multiple attempts. For comparison, we evaluate Nano Banana and also show Gemini 2.5 Pro's performance on mazes presented as images (I2T) or ASCII text (T2T).

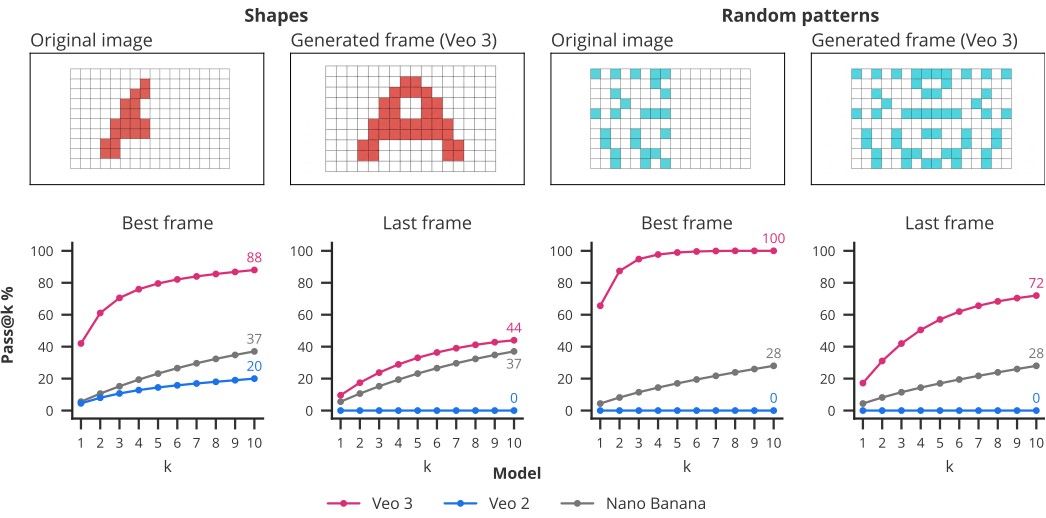

Figure 8: **Visual symmetry**. Prompt: *"Instantly reflect this pattern along the central, vertical axis while keeping the existing colored pattern without modification. [...]"* A model has to color all cells correctly to pass. Details & full prompt: App. B.6.

of a 2×2 grid to complete the analogy (e.g., A is to B as C is to ?). We evaluate the correctness of the generated infill for four transformation types from KiVA [70], see App. B.7 for details. The results are summarized in Fig. 9. While Veo 2 struggles to understand any analogies, Veo 3 correctly completes examples for *color* and *resize*. However, both models perform below chance (0.33) on *reflect* and *rotate* analogies, indicating an erroneous, systematic bias.

> **Takeaway 4**  While far from perfect, Veo 3—building on its ability to perceive, model and manipulate objects—shows emergent visual reasoning abilities.

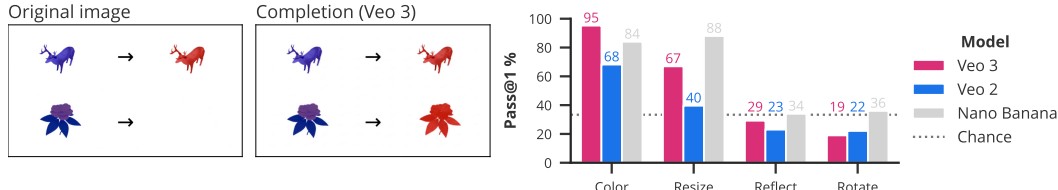

Figure 9: **Visual analogy solving** on four transformations à 50 samples from KiVA [70]. Prompt: *"[...] generate the missing object in the lower right region and solve the visual analogy. [...]"* Pass@1 is evaluated on the last frame, results up to pass@10 can be found in App. B.7.

## 5 DISCUSSION

**Summary** A breakthrough in machine vision started the deep learning revolution in 2012 [71], but in recent years, natural language processing has seen the most rapid progress. This was driven by the rise of general-purpose LLMs, whose ability to solve novel tasks in a zero-shot fashion has led them to replace most task-specific models in NLP. We here make the case that machine vision is on the cusp of a similar paradigm shift, enabled by emergent abilities of large-scale video models. Our core finding is that Veo 3 can solve a wide range of tasks in a zero-shot manner, spanning the full vision stack from **perception** to **modeling**, **manipulation** and even early forms of **visual reasoning**. While its performance is not yet perfect, the massive and consistent improvement from Veo 2 to Veo 3 indicates that video models will become general-purpose foundation models for vision, just as LLMs have for language.                                          NEW

**Zero-shot learning in the era of web-scale data** The advent of models trained on web-scale data has led to a necessary evolution in the understanding of zero-shot capabilities. Given the vast and diverse nature of the training data, it is difficult to verify that a model has not encountered data similar to a particular task before. In contemporary usage, the definition of "zero-shot learning" has therefore shifted from a strict "unseen task" criterion (often impossible to verify) to a methodological one which we adopt: A task is considered zero-shot if the model can be prompted to perform the task despite not being trained, adapted or fine-tuned specifically for that task [10, 30, 72]. For     NEW instance, LLMs were described as "zero-shot reasoners" [10] not because of any guarantees that rea-     NEW soning-style text isn't part of the training data by accident, but because they can perform reasoning tasks without task-specific training or adaptation, simply via prompting alone.

**Performance is a lower bound** Tasks can be represented in a myriad of ways; a maze, for example, can be presented as a black-and-white grid, a video game, or a photorealistic scene, with the prompt requesting a solution in the form of a line, a moving object, or a glowing path. Moreover, visually, a maze could be represented as a black-and-white grid, a Pac-Man game, or a photorealistic top-down view of an apartment. This has three implications: First, prompt engineering—including the *visual* prompt a.k.a. starting frame—is as important for visual tasks as it is for LLMs (see also App. C and [73] for a discussion). Second, we must distinguish between a model's task performance and its underlying ability (i.e., competence) to solve that task [74, 75]. Third, as a consequence, the model performance reported here with a given visual and textual prompt should be considered a lower bound on the model's true capabilities. This also holds for the tasks that we report as failure cases in App. E, such as providing visual instructions to fold laundry (Fig. 76), planning to move a sofa between rooms separated by a small door (Fig. 77), or certain visual puzzles (Fig. 70).          FIX

**Video generation is expensive, but costs tend to fall** Generating a video is more expensive than running a bespoke, task-specific model (see also App. D). Yet, the economics of general-purpose models are on a predictable trajectory: Epoch AI [76] estimates that LLM inference costs are falling by a factor of $9\times$ to $900\times$ per year for a given performance level. In NLP, early generalist models were also considered prohibitively expensive ("GPT-3's size makes it challenging to deploy" [7, p. 8]). Nevertheless, rapidly falling inference costs, combined with the appeal of generalist models, have replaced most task-specific language models. If NLP is a guide, the same trend will play out in vision. While specialized models will likely remain cheaper, the superior performance and poten-     NEW

tial cross-task synergies of foundation models increasingly justify their higher computational cost. For example, today's leading coding models aren't task-specific models, but general-purpose LLMs. Similarly, unified vision models will likely outperform many task-specific models since visual tasks have deep relationships with each other [16], enabling generalist models to benefit from synergies.

**Jack of many trades, master of few?** For many tasks, Veo 3's performance is below state of the art of specialized models. This mirrors the early days of LLMs; GPT-3 reported performance well below fine-tuned models on many tasks [7, cf. Tables 3.1, 3.3, 3.4, 3.5]). This did not stop language models from becoming foundation models, and we don't believe it will stop video models from becoming vision foundation models for two reasons. First, the step-change in performance from Veo 2 to Veo 3 is evidence of rapid progress over time. Second, our scaling results from Sec. 4 show pass@10 to be consistently higher than pass@1 with no signs of a plateau. Therefore, inference-time scaling methods [e.g. 77–80] in combination with the standard optimization toolkit like post-training with automatic verifiers are likely to boost performance. For the tasks we test here, Veo 3 is akin to a pre-trained language model that has yet to undergo instruction tuning or RLHF [81, 82].

**Will other video models show similar behavior?** While tackling visual intelligence tasks with video models is still in its infancy, three articles released after the initial version of this work confirm that other model families beyond Veo also show strong performance: [83] investigate Sora-2, [84] test six models including Sora-2, Hailuo-02 and Kling-2.5-Turbo-Pro, while [85] adapt CogVideoX1.5-5B for visual reasoning tasks. Collectively, they indicate that our finding of emergent visual intelligence in video models isn't limited to a single model family.

**Where does Veo's performance come from?** Veo's architecture, training data mixture, and training recipe, alongside the exact models used for embeddings and prompt-rewriting, and the specific verbal and visual prompts used in each task all likely influence the performance reported in this work. Related works published after the initial version of this manuscript and mentioned in the paragraph above indicate that many of the behaviors we demonstrate are not specific to Veo. Instead, they seem to be an emergent property of video generation trained on large-scale data. As is the case for large language models and large vision models [86–89], the exact performance distribution across tasks is likely determined by the data mixture, though more research is required. It also remains to be seen whether scaling laws can be established for visual intelligence tasks of video models, that properly capture the impact of data scale and compute similarly to prior works in LLMs [90–92]. On the prompt side, we examine the impact of the text prompt in App. C; a study on the impact of visual prompts and the potential for automatic prompt engineering is left for future work.

**Similarities and differences with NLP** There are strong reasons to believe that progress in vision will mirror the rapid advancements seen in NLP. Both language and vision models are developed on the same hardware infrastructure, and thus benefit from the same hardware scaling laws. Additionally, many pivotal algorithmic breakthroughs, such as the Transformer architecture, are modality-agnostic and have been successfully adapted for both domains. We therefore expect inference costs for vision models to decrease just as they are for LLMs, enabling their widespread adoption.

However, there are also crucial differences. A significant hurdle for vision is the cost of training on information-dense video data, which could slow the adoption of large-scale vision models compared to their NLP counterparts. Conversely, data availability presents a long-term advantage for vision. It is widely believed that we are approaching the limits of available high-quality text data, whereas the amount of video data being generated globally continues to grow exponentially. This suggests that while training costs are a near-term challenge, the vast and growing reservoir of video data may ultimately fuel the development of even more powerful foundation models for vision.

**Outlook** This is an exciting time for vision. Seeing NLP's recent transformation from task-specific to generalist models, it is conceivable that the same transformation will happen in machine vision through video models (a "GPT-3 moment for vision"), enabled by their emergent ability to perform a broad variety of tasks in a zero-shot fashion, from perception to visual reasoning.

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

# APPENDIX

## A    QUALITATIVE RESULTS:
## PERCEPTION, MODELING, MANIPULATION, REASONING

NEW

We show the initial prompt frame and successive generated frames for each task described in Sec. 3. For each task, the figure caption also states the prompt used to generate all samples. The guidelines used to determine success in each task are listed in App. A.5.

NEW

### A.1    PERCEPTION

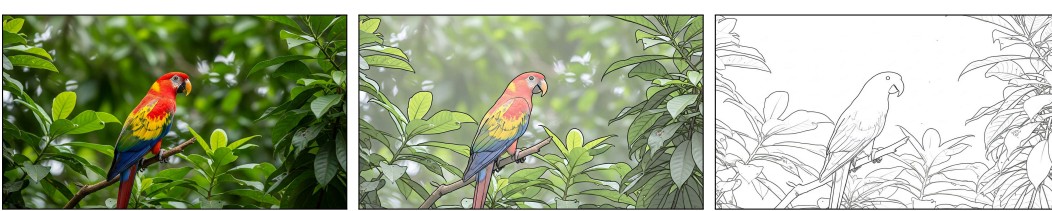

Figure 10: **Edge detection**. Prompt: *"All edges in this image become more salient by transforming into black outlines. Then, all objects fade away, with just the edges remaining on a white background. Static camera perspective, no zoom or pan."* Success rate: 0.92.

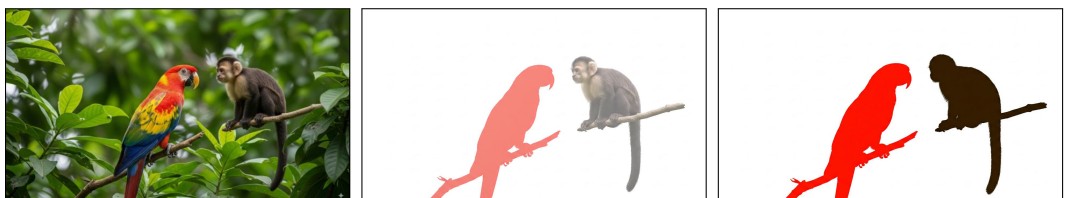

Figure 11: **Segmentation**. Prompt: *"Create an animation of instance segmentation being performed on this photograph: each distinct entity is overlaid in a different flat color [...]"* (full prompt: App. B.2). Success rate: 0.33.

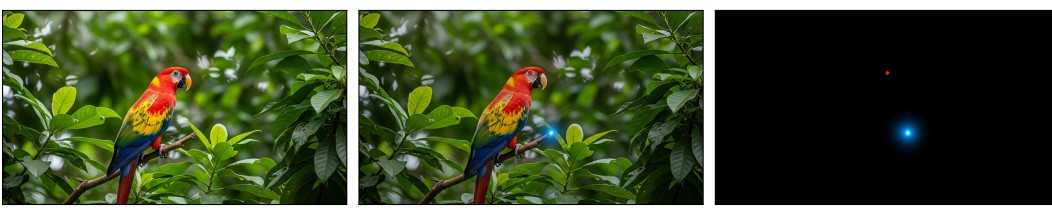

Figure 12: **Keypoint localization**. Prompt: *"Add a bright blue dot at the tip of the branch on which the macaw is sitting. The macaw's eye turns bright red. Everything else turns pitch black. Static camera perspective, no zoom or pan."* Success rate: 0.58.

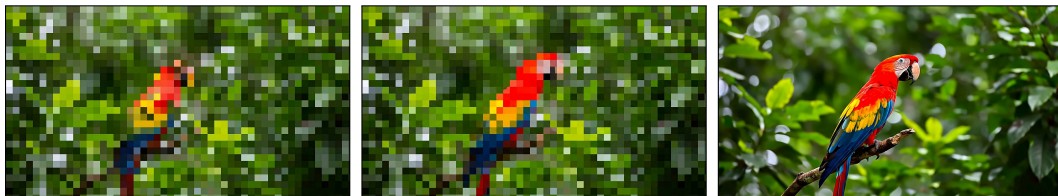

Figure 13: **Super-resolution**. Prompt: *"Perform superresolution on this image. Static camera perspective, no zoom or pan."* Success rate: 0.75.

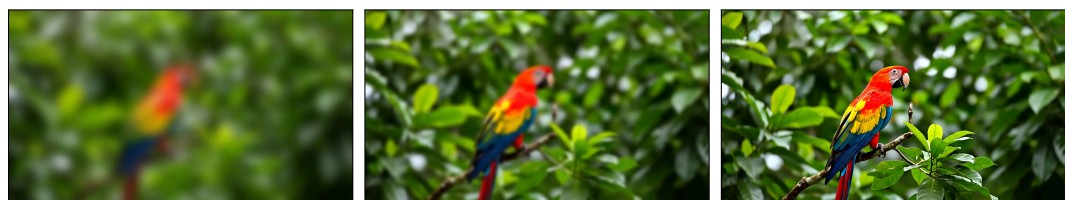

Figure 14: **Blind deblurring**. Prompt: *"Unblur image including background. Static camera perspective, no zoom or pan."* Success rate: 1.0.

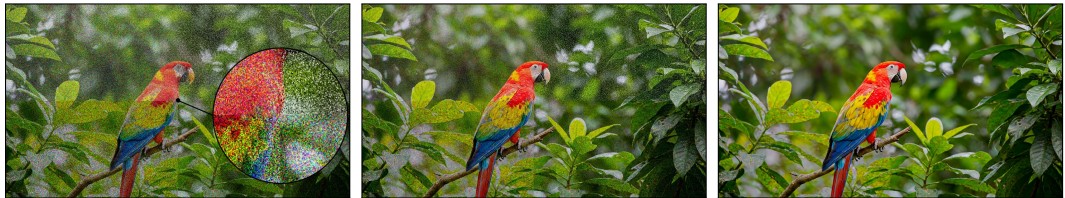

Figure 15: **Blind denoising**. Each quadrant was corrupted with a different type of noise. Clockwise from top left: Gaussian noise, salt-and-pepper noise, speckle noise, shot noise. Prompt: *"Remove the noise from this image. Static camera perspective, no zoom or pan."* Success rate: 1.0.

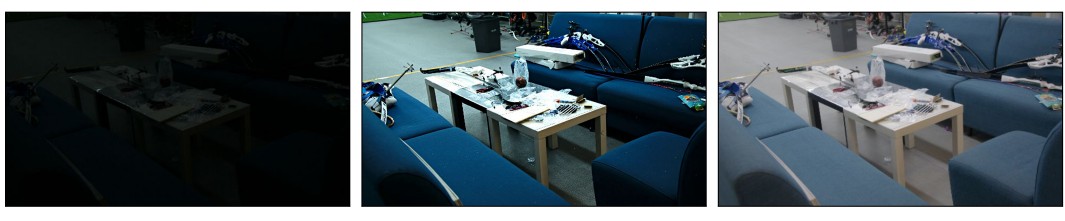

Original low-light image      Veo 3-generated lit image      Ground-truth lit image

Figure 16: **Low-light enhancing**. Prompt: *"Fully restore the light in this image. Static camera perspective, no zoom or pan."* Success rate: 0.92. Image and ground-truth source: LOLv2 dataset [93].

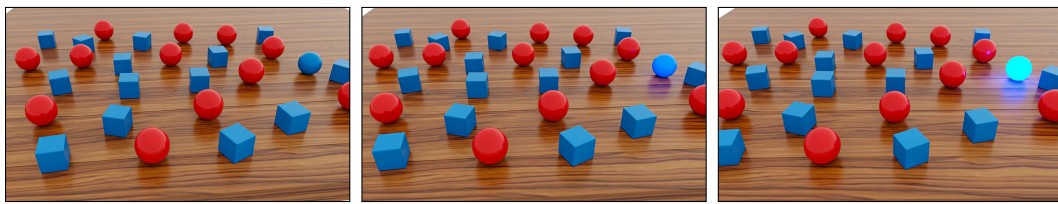

Figure 17: **Conjunctive search / binding problem**. Prompt: *"The blue ball instantly begins to glow. Static camera perspective, no zoom no pan no movement no dolly no rotation."* Success rate: 0.75. Inspiration: [94].

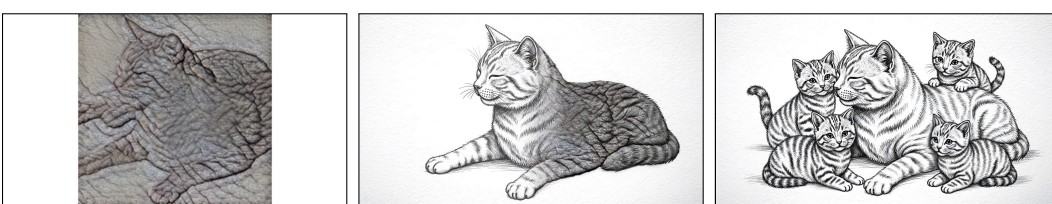

Figure 18: **Dalmatian illusion understanding**. Prompt: *"Static camera perspective."* Success rate: 1.0. Image credit: [95, 96].

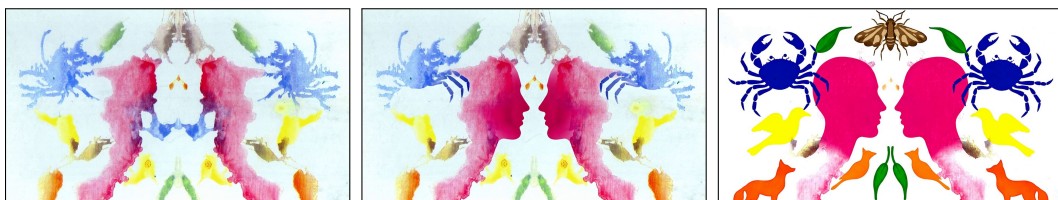

Figure 19: **Shape (cue-conflict) understanding**. Prompt: *"Transform the animal in this image into a sketch of the animal surrounded by its family."* Success rate: 1.0. Image credit: [97].

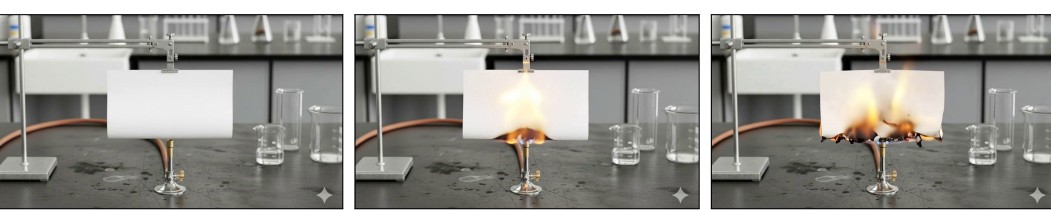

Figure 20: **Rorschach blot interpretation**. Prompt: *"The patterns transform into objects."* Success rate: undefined (1.0 for prompt following). Image credit: H. Rorschach, public domain via wikipedia.

## A.2 MODELING

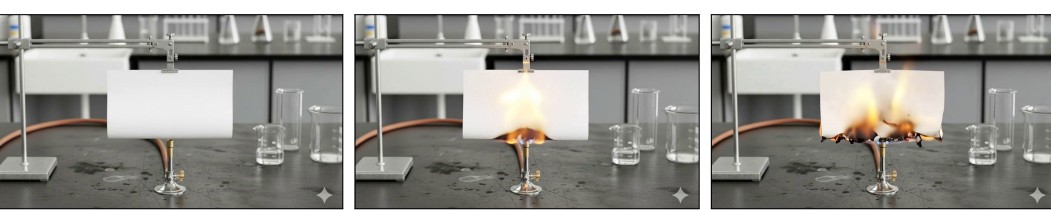

Figure 21: **Material properties.** Prompt: *"The bunsen burner at the bottom turns on. Sped up time lapse. Static camera, no pan, no zoom, no dolly."* Success rate: 0.25.

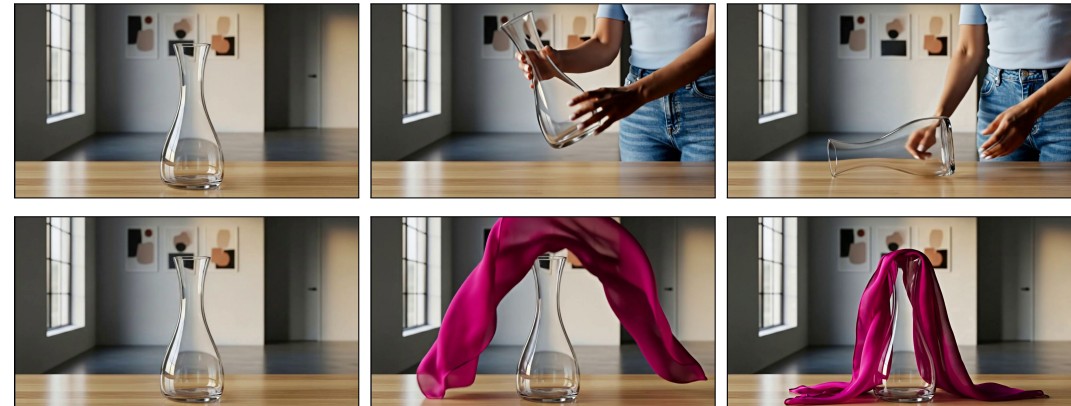

Figure 22: **Physics body transform**. **Rigid body** (top). Prompt: *"A person picks up the vase and puts it back on the table in a sideways orientation. Static camera, no pan, no zoom, no dolly."* Success rate: 1.0. **Soft body** (bottom). Prompt: *"A person drapes a thin silk scarf over the vase. Static camera, no pan, no zoom, no dolly."* Success rate: 0.67.

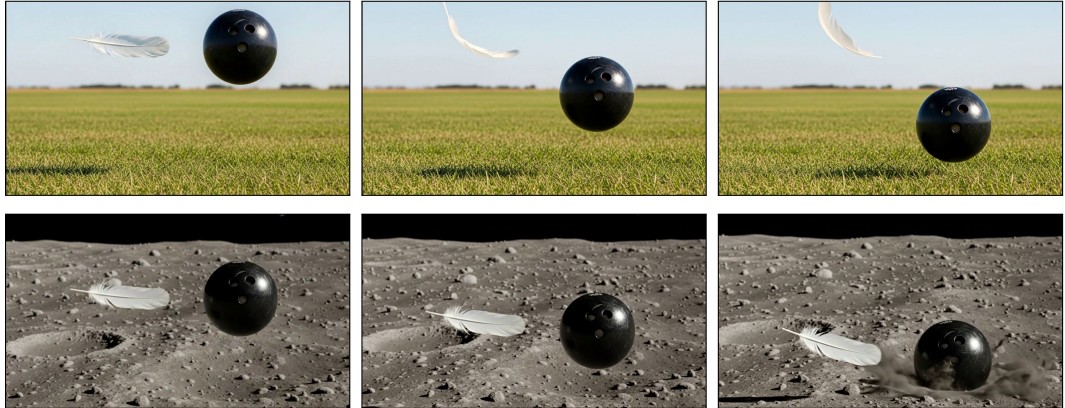

Figure 23: **Gravity and air resistance**. **On earth** (top). Prompt: *"The objects fall due to gravity. Static camera, no pan, no zoom, no dolly."* Success rate: 0.5. **On the moon** (bottom). Prompt: *"The objects fall down on the moon due to gravity. Static camera, no pan, no zoom, no dolly."* Success rate: 0.5.

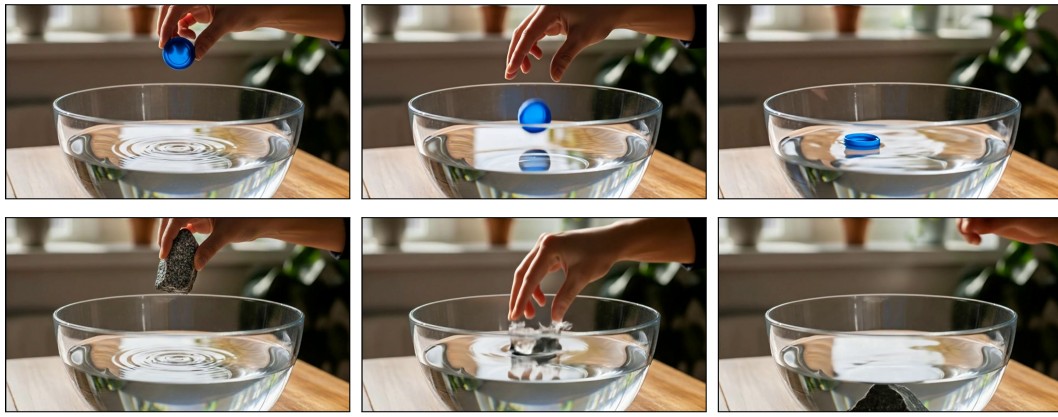

Figure 24: **Buoyancy.** Prompt: *"The hand lets go of the object. Static camera, no pan, no zoom, no dolly."* Success rate (bottle cap): 0.58; success rate (rock): 0.83.

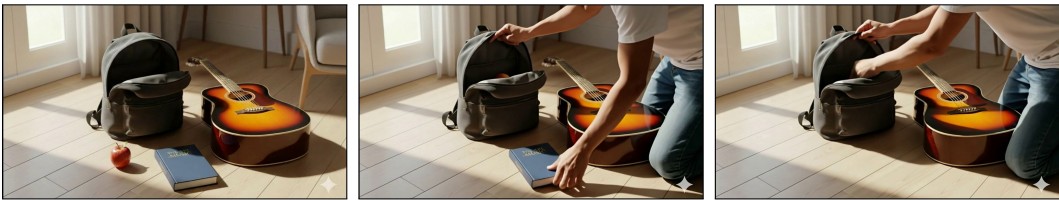

Figure 25: **Visual Jenga**, inspired by [55]. Prompt: *"A hand quickly removes each of the items in this image, one at a time."* Success rate, based on removal of at least three objects: 0.5.

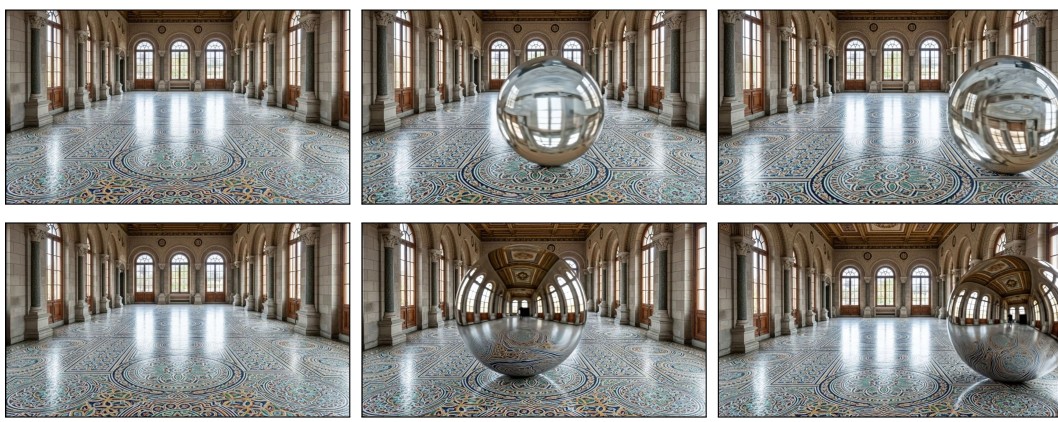

Figure 26: **Object packing**. Prompt: *"A person puts all the objects that can fit in the backpack inside of it. Static camera, no pan, no zoom, no dolly."* Success rate: 0.75.

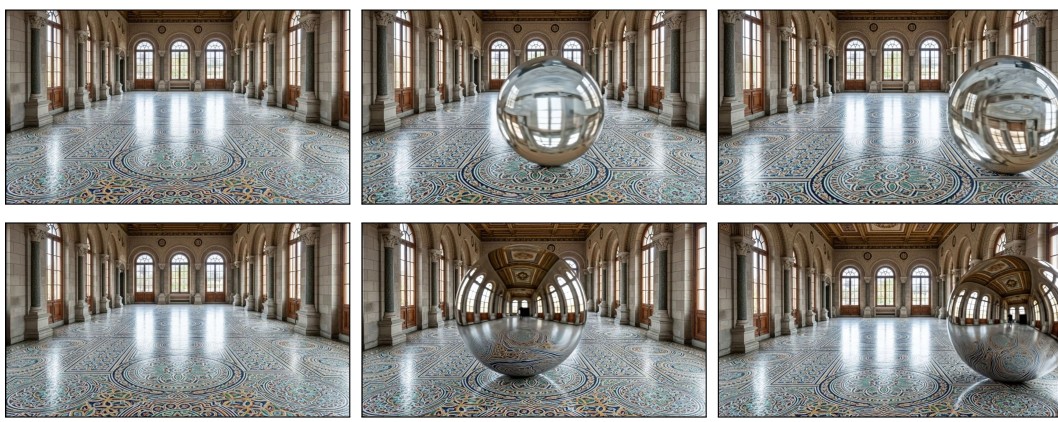

Figure 27: **Material optics**. **Glass** (top). Prompt: *"A giant glass sphere rolls through the room. Static camera, no pan, no zoom, no dolly."* Note that the image through the glass sphere is inverted. Success rate: 0.92. **Mirror** (bottom). Prompt: *"A giant mirror-polish metal sphere rolls through the room. Static camera, no pan, no zoom, no dolly."* Note that the image reflected off the sphere is not inverted). Success rate: 1.0.

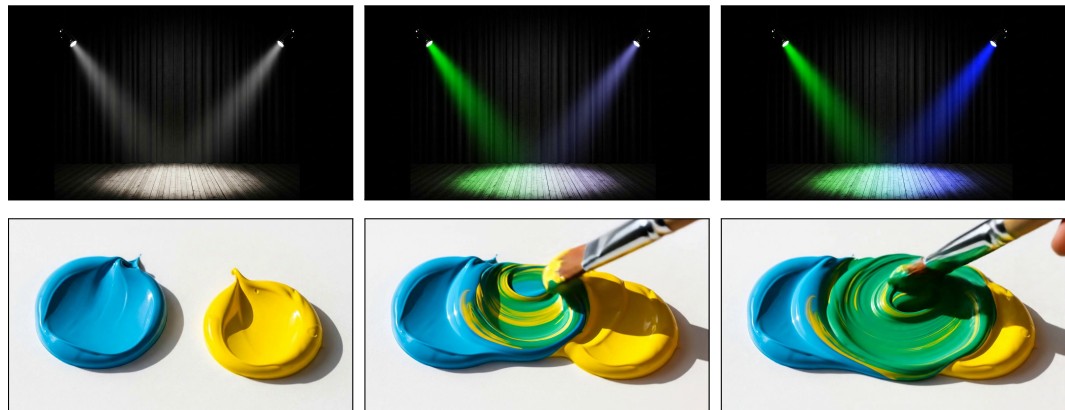

Figure 28: **Color mixing**. **Additive** (lights, top). Prompt: *"The spotlight on the left changes color to green, and the spotlight on the right changes color to blue."* Success rate: 0.92. **Subtractive** (paints, bottom). Prompt: *"A paintbrush mixes these colors together thoroughly until they blend completely. Static camera, no pan, no zoom."* Success rate: 0.75.

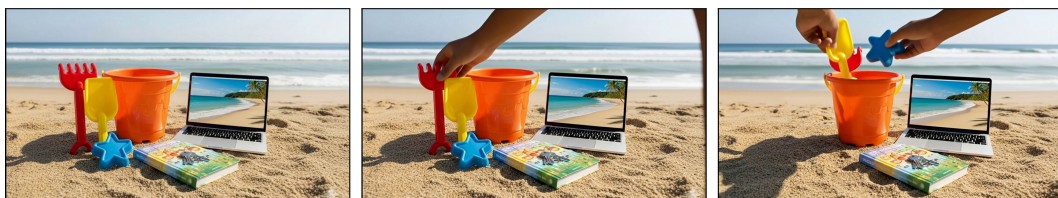

Figure 29: **Categorizing objects.** Prompt: *"A person puts all the kids toys in the bucket. Static camera, no pan, no zoom, no dolly."* Success rate: 0.33.

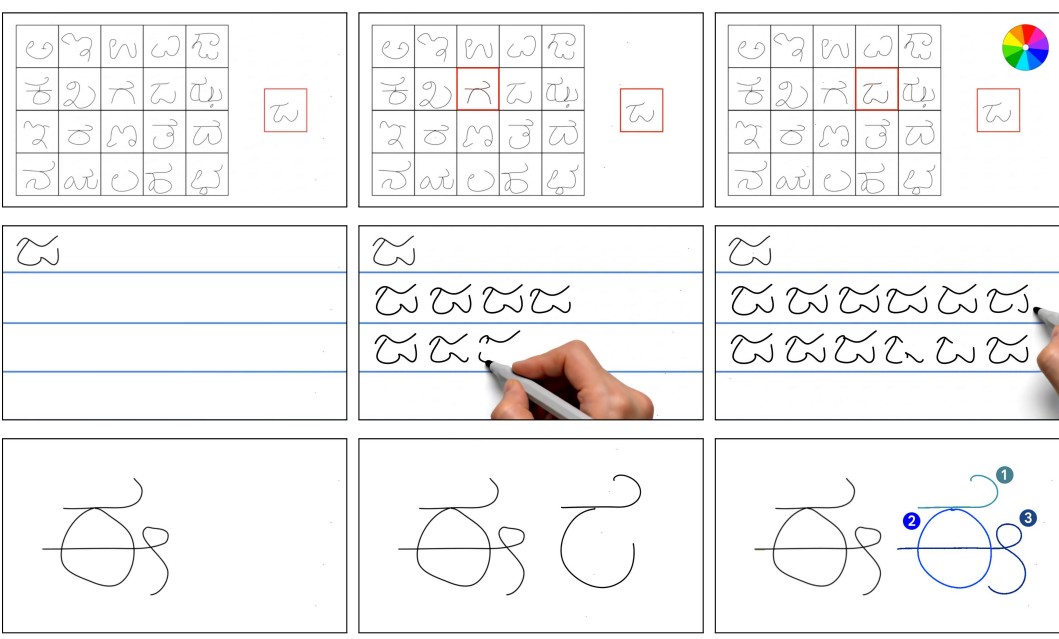

Figure 30: **Character recognition, generation, and parsing**, inspired by the Omniglot dataset [56]. **Recognition** (top). Prompt: *"The background of the grid cell with the same symbol as the one indicated on the right turns red. All other grid cells remain unchanged. After that, a spinning color wheel appears in the top right corner."* (Note: Veo 3 has a prior to keep things moving, which is detrimental for tasks where the solution is obtained in an early frame. We observe that a 'motion outlet', such as a color wheel, can indicate task completion and 'freeze' the solution.) Success rate: 0.33. **Generation of variations** (middle). Prompt: *"The page is filled line-by-line with hand-written practice variations of the symbol."* Success rate: 0.25. **Parsing into parts** (bottom). Color and numbers in final frame are added post-hoc to show stroke order. Prompt: *"Stroke-by-stroke, a replica of the symbol is drawn on the right."* Success rate: 0.5.

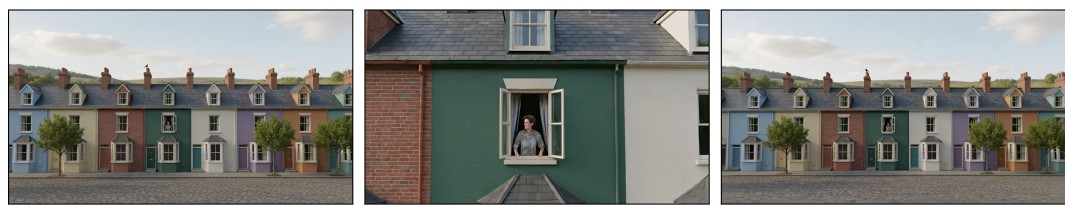

Figure 31: **Memory of world states**. Prompt: *"The camera zooms in to give a close up of the person looking out the window, then zooms back out to return to the original view."* Success rate: 1.0.

## A.3 MANIPULATION

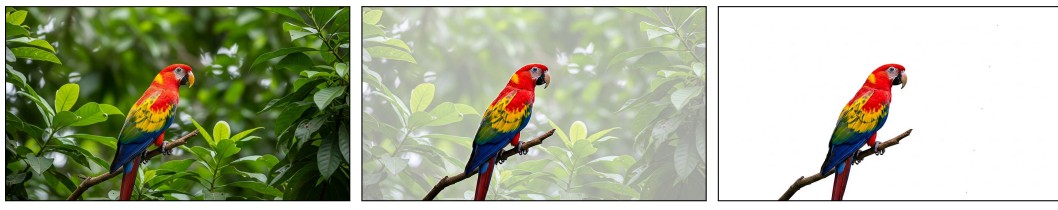

Figure 32: **Background removal**. Prompt: *"The background changes to white. Static camera perspective, no zoom or pan."* Success rate: 0.83.

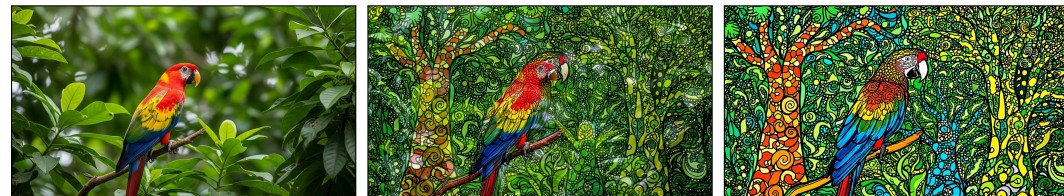

Figure 33: **Style transfer**. Prompt: *"The scene transforms into the style of a Hundertwasser painting, without changing perspective or orientation; the macaw does not move. Static camera perspective, no zoom or pan."* Success rate: 0.75.

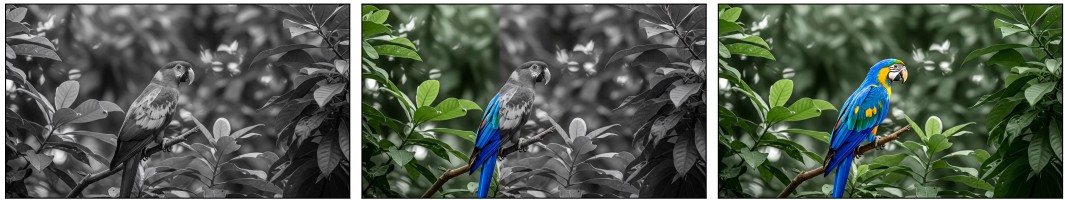

Figure 34: **Colorization**. Prompt: *"Perform colorization on this image. Static camera perspective, no zoom or pan."* Success rate: 0.08.

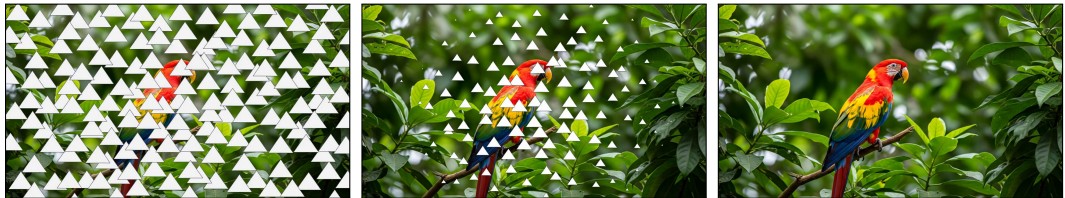

Figure 35: **Inpainting**. Prompt: *"The white triangles become smaller and smaller, then disappear altogether. Static camera perspective, no zoom or pan."* Success rate: 1.0.

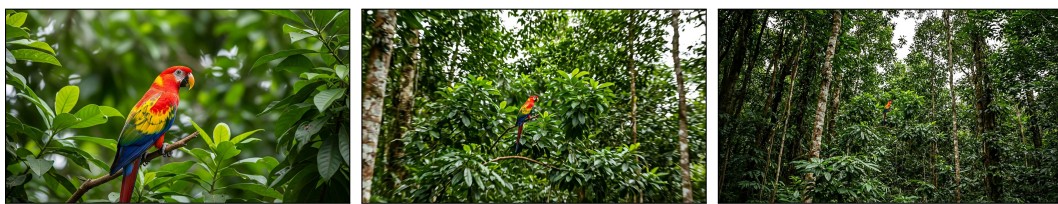

Figure 36: **Outpainting**. Prompt: *"Rapidly zoom out of this static image, revealing what's around it. The camera just zooms back, while the scene itself and everything in it does not move or change at all, it's a static image."* Success rate: 1.0.

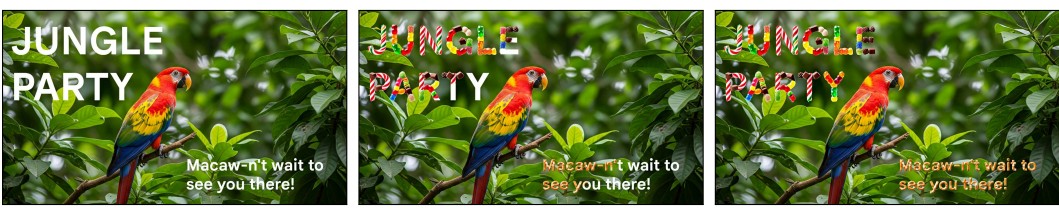

Figure 37: **Text manipulation**. Prompt: *"Animation of the text rapidly changing so that it is made out of different types of candy (top left text) and pretzel sticks (bottom right text). Static camera perspective, no zoom or pan."* Success rate: 0.33.

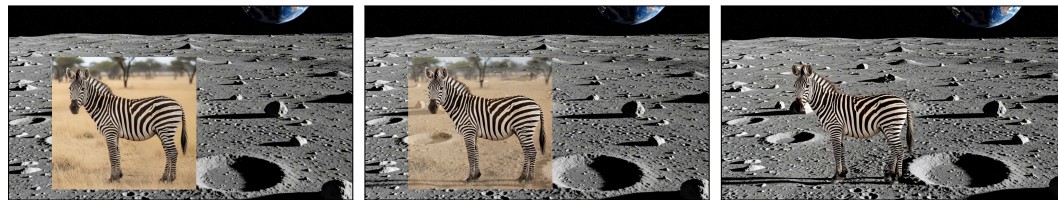

Figure 38: **Image editing with doodles**. Prompt: *"Changes happen instantly."* Success rate: 1.0.



Figure 39: **Scene composition**. Prompt: *"A smooth animation blends the zebra naturally into the scene, removing the background of the zebra image, so that the angle, lighting, and shading look realistic. The final scene perfectly incorporates the zebra into the scene."* Success rate: 0.75.

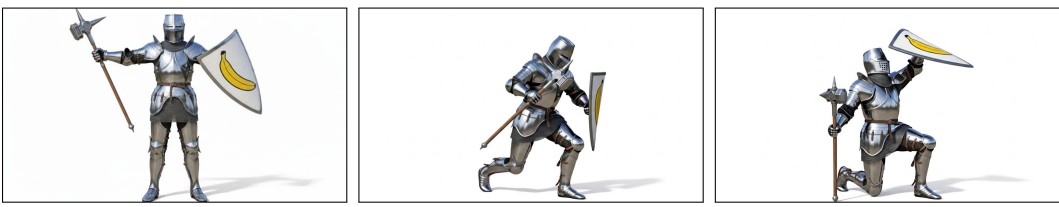

Figure 40: **Single-image novel view synthesis**. Prompt: *"Create a smooth, realistic animation where the camera seems to rotate around the object showing the object from all the sides. Do not change anything else. No zoom. No pan."* Success rate: 0.92. Image source: [98].

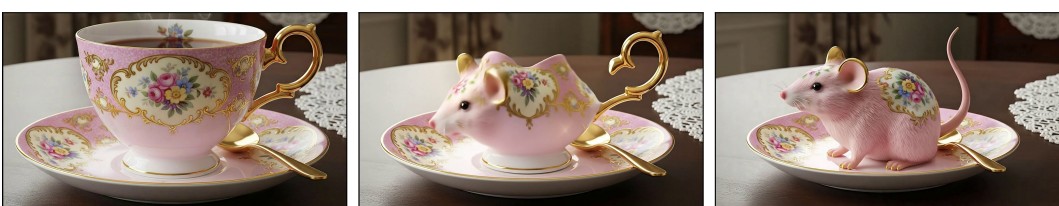

Figure 41: **3D-aware reposing**. Prompt: *"The knight turns to face to the right and drops on one knee, lifting the shield above his head to protect himself and resting the hilt of his weapon on the ground."* Success rate: 0.83.

Figure 42: **Transfiguration**. Prompt: *"A magical spell smoothly transforms the structure of the teacup into a mouse."* Success rate: 0.17.

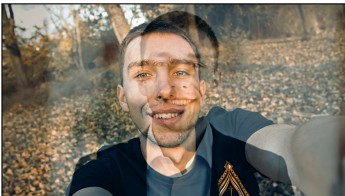 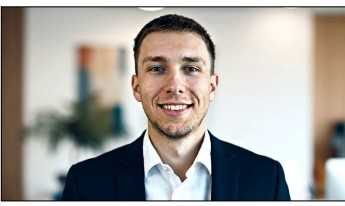

Figure 43: **Professional headshot generation**. Prompt: *"Turn this selfie into a professional head-shot for LinkedIn."* Success rate: 0.42. Image credit: photo by George Pisarevsky on Unsplash.

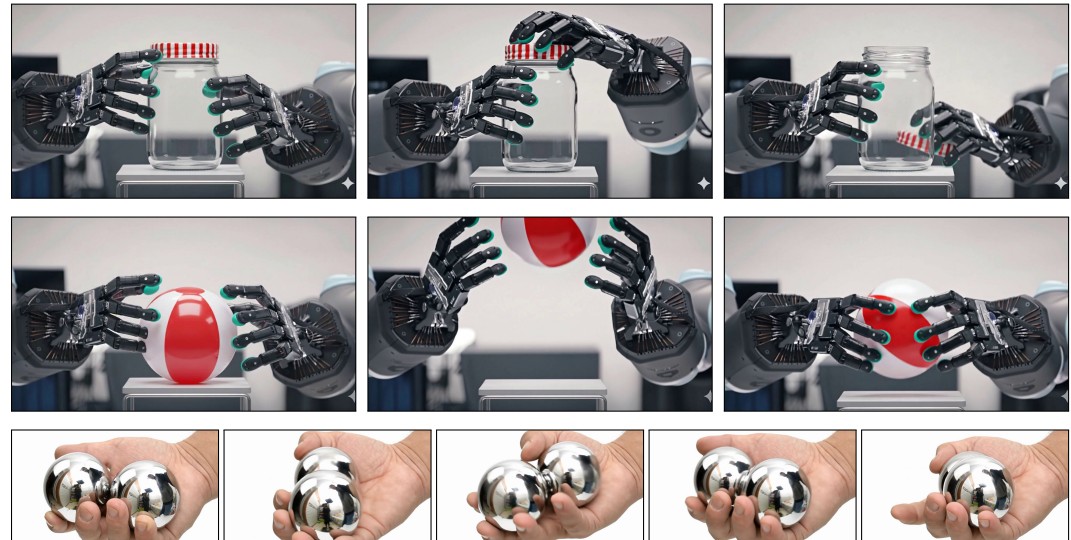

Figure 44: **Dexterous manipulation**. **Jar opening** (top). Prompt: *"Use common sense and have the two robot hands attached to robot arms open the jar, like how a human would."* Success rate: 1.0. **Throwing and catching** (middle). Prompt: *"Use common sense and have the two robot hands attached to robot arms throw the ball in the air, the ball goes up off the screen, hands move to positions to catch the ball, and catch the falling ball, like how a human would."* Success rate: 1.0. **Rotating Baoding balls** (bottom). Prompt: *"A human hand holds two metal Baoding balls. The fingers, including the thumb, index, and middle finger, skillfully manipulate the balls, causing them to rotate smoothly like two planets orbiting around each other and continuously in the palm, one ball circling the other in a fluid motion."* Success rate: 0.08.

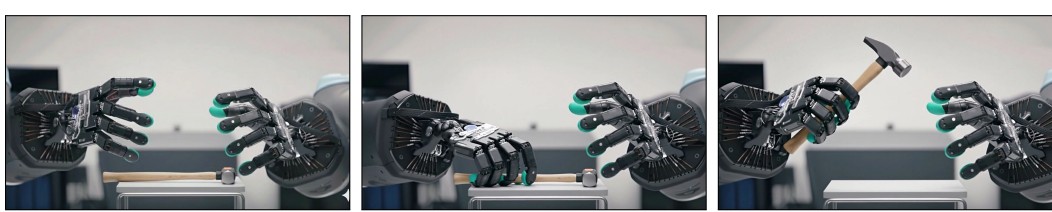

Figure 45: **Affordance recognition**. Prompt: *"The robot hands mounted on robot arms pick up the hammer, naturally like how a human would."* Success rate: 0.5.

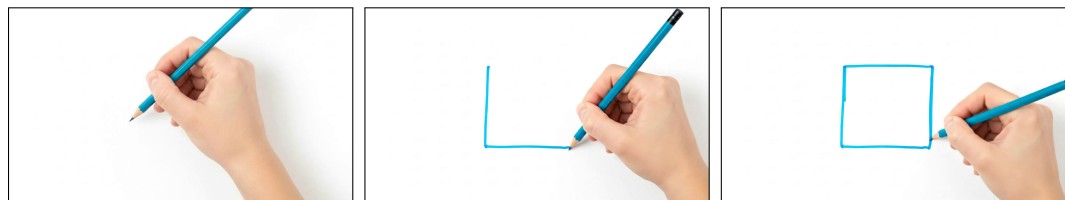

Figure 46: **Drawing**. Prompt: *"A person draws a square. Static camera, no pan, no zoom, no dolly."* Success rate: 0.33.

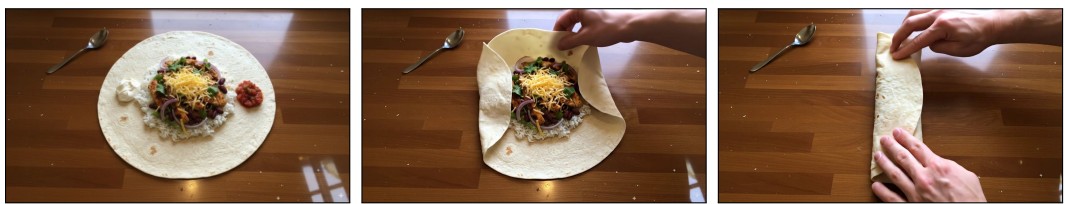

Figure 47: **Visual instruction generation**. Prompt: *"A montage clearly showing each step to roll a burrito."* Success rate: 0.25. Inspiration: [27] and Reddit.

## A.4 REASONING

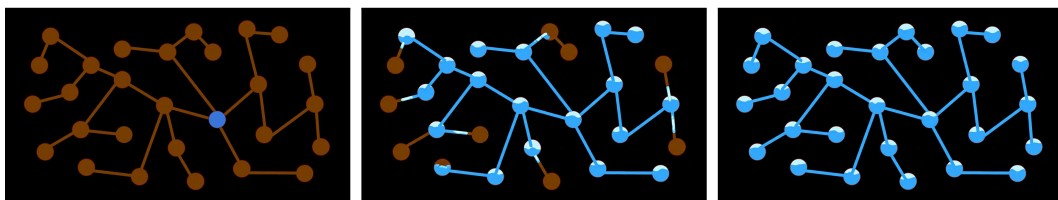

Figure 48: **Graph traversal**. Prompt: *"Starting from the blue well, an unlimited supply of blue water moves through the connected channel system without spilling into the black area."* Success rate: 0.08.

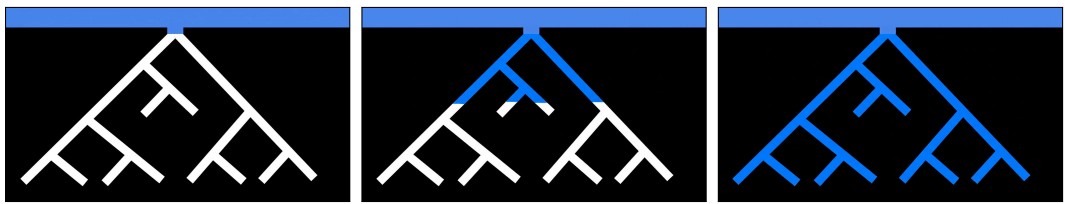

Figure 49: **Tree BFS**. Prompt: *"From the blue water basin, an unlimited supply of water flows at constant speed into the cave system until all caves are filled. Static camera perspective, no zoom or pan."* Success rate: 0.17.

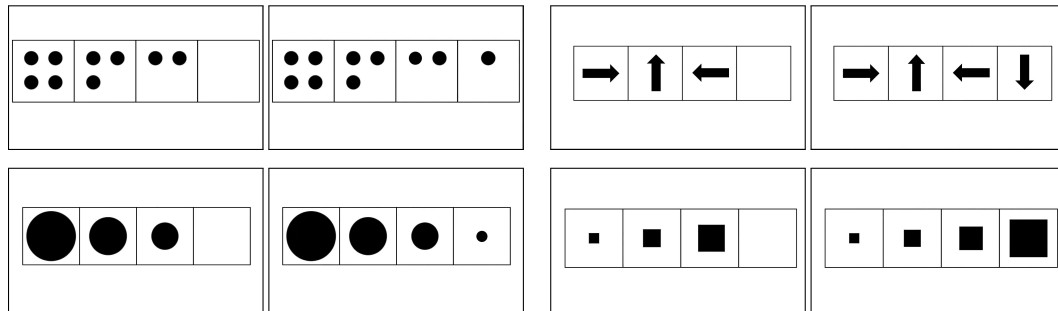

Figure 50: **Sequence completion** inspired by Raven's progressive matrices. Each of the four pairs shows input (left) and generated output (right). Prompt: *"Draw the figure that completes the pattern in the rightmost box. The images in the boxes are static. Do not modify the existing images, only draw in the empty box. Static camera, no zoom, no pan, no dolly."* Success rate: 0.33 for dots, 1.0 for arrows, 0.75 for shrinking circles, 0.83 for growing squares.

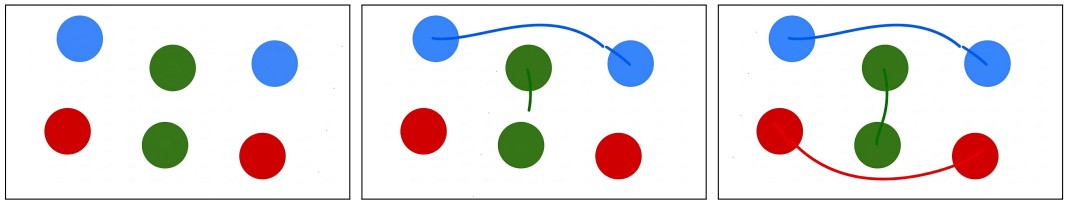

Figure 51: **Connecting colors**. Prompt: *"Draw three curves, one connecting each pair of circles of the same color."* Success rate: 0.25.

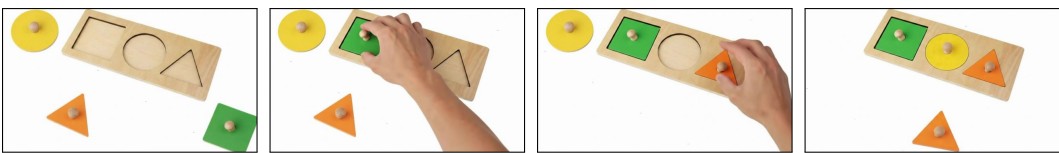

Figure 52: **Shape fitting**. Prompt: *"The scene shows three colored pieces, and a wooden panel with three holes. Each colored piece fits into one and only one hole. A hand grabs each colored piece and puts it into an empty hole that has the exact same shape - if it doesn't fit, the hand tries another hole. All the objects must be placed in their respective holes."* Success rate: 0.25.

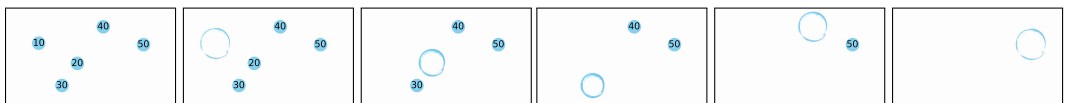

Figure 53: **Sorting numbers**. Prompt: *"The video starts with some numbered bubbles. The bubbles pop and disappear one at a time, in numeric order, starting from the one with the smallest number."* Success rate: 0.08.

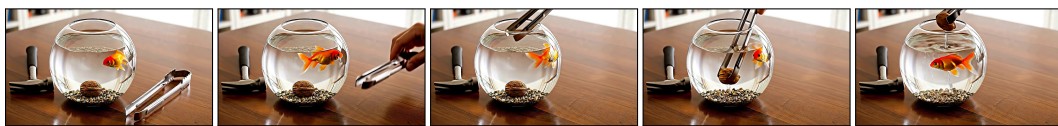

Figure 54: **Tool use**. Prompt: *"A person retrieves the walnut from the aquarium."* Success rate: 0.92 (retrieval via tool) and 0.08 (retrieval via tool without intersecting the glass).

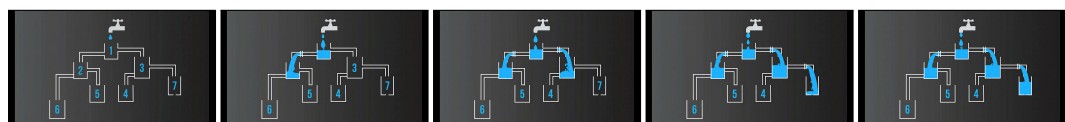

Figure 55: **Simple Sudoku completion**. Prompt: *"Create a static, smooth, animation that solves the given 4x4 sudoku. Enter the missing numbers one by one. Do not change anything else in the picture. Only fill the numbers in the empty cells so the sudoku is solved properly. A cursor moves and fills the correct number in the empty boxes."* Success rate: 0.67.

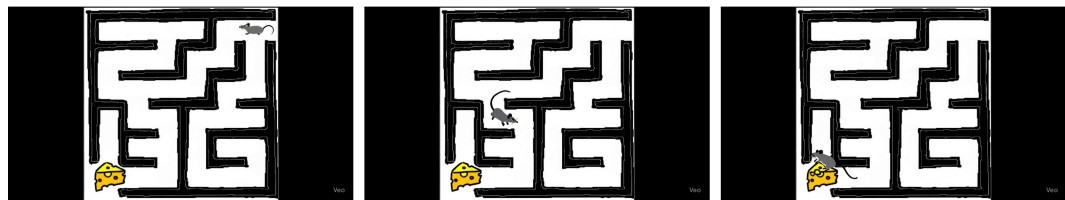

Figure 56: **Water puzzle solving**. Prompt: *"The tap is turned on and water starts flowing rapidly filling the containers. Create a smooth, static animation showing the containers getting filled with water in the correct order."* (note: not all containers can be filled since some pipes are closed off, such as the pipe connecting container 2 to container 5. Veo fills the correct containers, in the right order.) Success rate: 0.5.

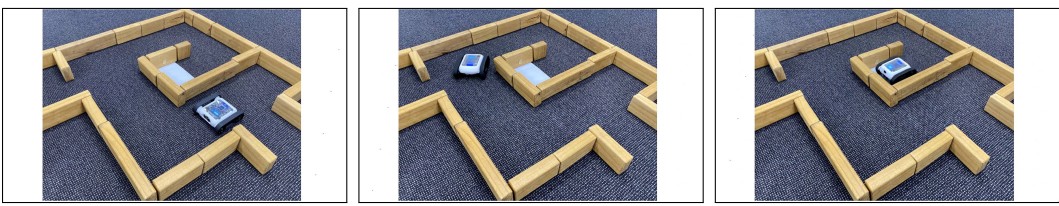

Figure 57: **Maze solving**. Prompt: *"Without crossing any black boundary, the grey mouse from the corner skillfully navigates the maze by walking around until it finds the yellow cheese."* Success rate: 0.17.

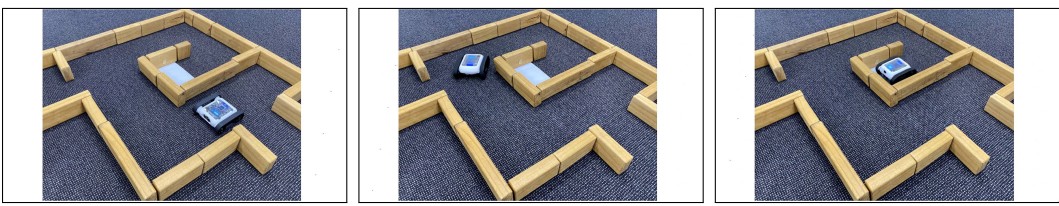

Figure 58: **Robot navigation**. Prompt: *"The robot drives to the blue area. Static camera perspective, no movement no zoom no scan no pan."* Success rate: 0.58. Image credit: Micromelon Robotics website with permission from Tim Hadwen.

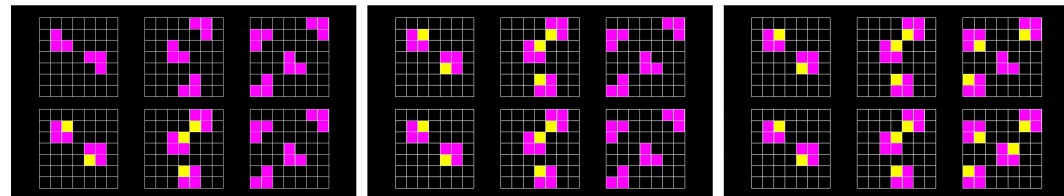

Figure 59: **Rule extrapolation** inspired by ARC-AGI [99]. Prompt: *"Modify the lower-right grid to adhere to the rule established by the other grids. You can fill cells, clear cells, or change a cell's color. Only modify the lower-right grid, don't modify any of the other grids. Static scene, no zoom, no pan, no dolly."* Success rate: 0.08. While Veo 3 doesn't follow the prompt perfectly, the output grid (bottom right) is completed correctly.

## A.5 EVALUATION DETAILS

NEW

For each task, we generate 22 completions with Veo 3 based on the same prompt and initial frame. Twelve completions are annotated by one of the authors, another ten samples are each annotated by two authors and one external computer vision expert. All annotations follow the general and specific instructions outlined below. The final success rate is computed as the average success over all samples, where the success rate for a sample with multiple annotations is the average of its annotations.

NEW

We track the pair-wise inter-rater reliability between annotators for the ten samples using Cohen's kappa [100]. $\kappa$ is 0.82 between authors and 0.87 and 0.89 between each author and the external annotator, compared to an analytical upper bound of 1.0 [101]. This is considered *strong* to *almost perfect* agreement [100, 102], indicating a stable and objective evaluation.

NEW
NEW
NEW
NEW

Annotators were given the following general instructions:

- Indicate whether the generation fulfills the task outlined in the prompt (1) or not (0).
- For tasks where the target is a single image (e.g., edge detection, deblurring, ...), indicate 1 if there is at least one frame that fulfills the task.
- For tasks where the target is a video segment (e.g., robot navigation, ...), indicate 1 if there is a video segment that fulfills the task.
- After the task has been completed, subsequent hallucinations or other artifacts are okay and don't invalidate the answer.
- While prompts may specify a static camera perspective, small camera movements are okay (e.g., zoom, pan, ...).
- Hallucinations or glitches are okay if they do not interfere with the task objective.
- Very small animations or local changes are okay if the task is otherwise fulfilled correctly.
- You can ignore any audio.

NEW

Additionally, annotators had the following specific instructions for some tasks

- **Background removal**. Some branches can be considered background or foreground.
- **Buoyancy (bottle cap)**. Does the bottle cap float up after being submerged?
- **Buoyancy (rock)**. Does the stone sink?
- **Categorizing objects**. Are all and only kids toys put in the bucket?
- **Color mixing (additive)**. Is the overlapping light cyan?
- **Color mixing (subtractive)**. Is the mixed paint green?
- **Dalmation illusion understanding**. Does the video reveal the Dalmatian?
- **Deblurring**. It is okay for the background to remain slightly out of focus.
- **Dexterous manipulation (jar opening)**. The entire interaction should be plausible.

- **Dexterous manipulation (throwing and catching)**. The entire interaction should be plausible.

- **Image editing with doodles**. Ignoring the speed at which the changes happen, are the changes that are introduced the requested ones (hat, snow, scarf)?

- **Drawing**. Is there a complete square, and each line is drawn with the pen? Lines can be drawn or re-drawn in any order.

- **Edge detection**. Does the model produce a plausible edge-map of the scene?

- **Graph traversal**. Are the edges/nodes filled in in the correct order? Not all edges/nodes have to be filled in during the 8s video. Score as failure if water appears somewhere without flowing from the initial well.

- **Gravity and air resistance (on earth)**. Does the feather fall slower than the ball due to air resistance?

- **Gravity and air resistance (on the moon)**. Do the feather and the ball fall at the same rate?

- **Professional headshot generation**. Is the likeness of the person preserved?

- **Material properties**. Does the paper burn in a plausible manner?

- **Memory of world states**. Does the scene in the end match the scene in the beginning?

- **Single-image novel view synthesis**. It's fine if the rotation isn't 360 degrees.

- **Character recognition**. At the moment the spinning color wheel appears, is only the correct symbol highlighted?

- **Character generation of variations**. Are the new characters plausible, hand-written variations of the original?

- **Character parsing into parts**. Is the character decomposed into plausible strokes?

- **Material optics (glass)**. Is the image in the sphere inverted?

- **Material optics (mirror)**. Is the image in the sphere right-side up?

- **3D-aware reposing**. Is there a frame showing the knight in the correct final pose, while maintaining his appearance? Implausible transformations are okay as long as the final pose is correct.

- **Physics body transform (rigid body)**. Does the vase retain its shape and appearance?

- **Physics body transform (soft body)**. Does the scarf drape plausibly over the vase?

- **Robot navigation**. Does the robot drive to the blue area in a physically plausible manner? (If e.g. a wall is knocked over by the robot, that could be plausible as well, as long as it could happen in reality)

- **Rule extrapolation**. Is the lower-right grid completed correctly? Modifications to the other grids are okay.

- **Scene composition**. The zebra does not have to maintain the pose from the image, but should be included plausibly in the scene (e.g., not hovering above the ground).

- **Segmentation**. Does the model produce a plausible segmentation of some key objects in the scene, while not introducing new objects?

- **Sequence completion**. Only focusing on the rightmost box and ignoring potential changes elsewhere, is the correct shape drawn?

- **Shape cue-conflict understanding**. Does the video show a family of cats?

- **Shape fitting**. Is there a moment when all holes are filled with the correct shape?

- **Style transfer**. Does the style of the image change to a painting?

- **Simple sudoku completion**. Are the correct numbers filled in? Top-left: 3, top-right: 4, bottom-left: 1. It is okay if other numbers are modified.

- **Tool use**. Is the walnut retrieved from the aquarium in a plausible manner?

- **Tree BFS**. Are the branches filled in a plausible manner (top to bottom)? Not all branches have to be filled in during the 8s video.

- **Visual instruction generation**. Are all necessary steps shown? A time-lapse/montage is okay.

- **Visual jenga**. Are the objects removed in a plausible order? Weird hand movements or object deformations are okay as long the order is plausible and objects are not reintroduced.

- **Water puzzle solving**. Are only the correct containers filled in, and in the right order 1-2-3-7?

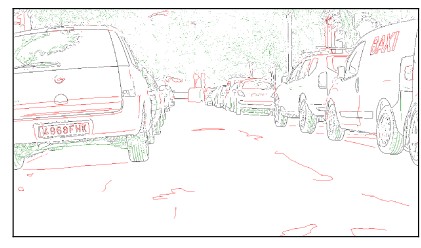

Figure 60: **Graded Veo 3 edge map**. While false negatives reflect genuine oversights of Veo 3 (e.g., cracks in the road, lettering on the car), many false positives correspond to actual image details that seem to be erroneously excluded from the ground truth (e.g., the outline of the trees, the reflection in the car window, and the tire profiles).

## B  QUANTITATIVE RESULTS: EXPERIMENTAL DETAILS

Table 1: **Video count breakdown for quantitative tasks**. For segmentation, $2 \times 1$ splits indicate one test set with two different background color prompts (white/green). For the prompt sensitivity study on the symmetry task (App. C), $2 \times 10$ splits indicate 2 splits (random/shape) across 10 tested prompt variations. In total, there are 884 different starting images for the quantitative tasks, and 46 for the qualitative tasks (if all macaw-based images are counted as a single image). For the qualitative tasks, we additionally generated 1364 videos (62 tasks $\times$ 22 samples).

| Task | Splits | Imgs/split | Pass@ | Video models | Total videos |
|---|---|---|---|---|---|
| Edge | 1 | 50 | 10 | 2 | 1000 |
| Segmentation | $2 \times 1$ | 50 | 10 | 2 | 2000 |
| Object extraction | 1 | 54 | 10 | 2 | 1080 |
| Editing | 1 | 30 | 1 | 2 | 60 |
| Maze | $2 \times 4$ | 50 | 10 | 2 | 8000 |
| Symmetry | 2 | 25 | 10 | 2 | 1000 |
| Symmetry prompt analysis | $2 \times 10$ | 25 | 1 | 1 | 500 |
| Analogy | 4 | 50 | 10 | 2 | 4000 |
| Total | | | | | 17640 |

### B.1  PERCEPTION: EDGE DETECTION

We provide details for the image editing task in Sec. 4.1.

**Evaluation**  As is standard in the literature, we refine and binarize predicted edges and allow for small local shifts compared to the ground truth [103–106]. Concretely, we use non-maximum suppression, then binarize with one of 16 evenly-spaced thresholds, then thin the binary edge map. At each threshold, we find the optimal mapping between predicted and ground-truth edge pixels within a radius of 0.75% of the image diagonal (around 11 pixels). Fig. 60 shows an example rating of a Veo 3-generated edge map. We report the best OIS over $k$ attempts (optimal image scale; the maximum $F_1$-score over all thresholds) for the best/last frame.

**Dataset**  We used all 50 test images from BIPEDv2 [64, 65].

**Models & prompts**  We tested Veo 3 `veo-3.0-generate-preview` and Veo 2 `veo-2.0-generate-preview-001` through the Vertex AI API. We also tested Nano Banana `gemini-2.5-flash-image-preview` through Google AI Studio.

> **Veo**
> *All edges in this image become more salient by transforming into black outlines. Then, all objects fade away, with just the edges remaining on a white background. Static camera perspective, no zoom or pan.*

> **Nano Banana**
> *Outline all edges in the image in black, make everything else white.*

**Sampling** We generated 10 videos per sample with a fixed prompt.

### B.2 PERCEPTION: SEGMENTATION

We provide details for the image editing task in Sec. 4.2.

**Evaluation** Since the model is free to choose any colors for segmentation masks, we first determine the number and hue of each mask by considering the hue-difference histogram between the original image and the extracted frame. We smooth the histogram with `scipy.ndimage.gaussian_filter1d` with a standard deviation of 2. Peaks with a minimum height of 10% of the maximum and at least 10 hue steps apart are considered to correspond to predicted segmentation masks. We then map each pixel to the mask with the closest hue.

Contrary to classic instance segmentation [66] or promptable segmentation [11, 12], our prompts do not specify a class or list of possible classes, a location prior (e.g., point or bounding box), or the number of instances in the image. This also means that mapping between predictions and annotated instances is established. Instead, we pair each ground-truth mask (including the background) with the predicted mask with the highest IoU (intersection over union), if any. We report mIoU as the average IoU over all pairs (excluding the background).

**Dataset** We evaluated on 50 randomly chosen test images from LVIS [66] that contain one to three objects, each with at least 5000 pixels.

**Models & prompts** We tested Veo 3 `veo-3.0-generate-preview` and Veo 2 `veo-2.0-generate-preview-001` through the Vertex AI API. We also tested Nano Banana `gemini-2.5-flash-image-preview` through Google AI Studio.

**Veo**
*Create an animation of instance segmentation being performed on this photograph: each distinct entity is overlaid in a different flat color.*

*Scene:*

- *The animation starts from the provided, unaltered photograph.*
- *The scene in the photograph is static and doesn't move.*
- *First, the background fades to {white, green}.*
- *Then, the first entity is covered by a flat color, perfectly preserving its silhouette.*
- *Then the second entity, too, is covered by a different flat color, perfectly preserving its silhouette.*
- *One by one, each entity is covered by a different flat color.*
- *Finally, all entities are covered with different colors.*

*Camera:*

- *Static shot without camera movement.*
- *No pan.*
- *No rotation.*
- *No zoom.*
- *No glitches or artifacts.*

**Nano Banana**
Perform instance segmentation on this image: Mask each distinct entity in a different opaque flat color that only preserves the silhouette and turn the background green.

**Sampling** We generated 10 videos per sample and prompt.

### B.3 MANIPULATION: OBJECT EXTRACTION

We provide details for the image editing task in Sec. 4.3.

**Evaluation** We extract the last frame from each generated video; the resulting image is converted to greyscale, a binary mask with threshold 200 is applied, and the number of connected components is extracted using `scipy.ndimage.label`, resulting in the count estimate. We also report the chance baseline which can be calculated as: $\mathsf{random - chance} = 1 - (1 - p)^k$ where $p$ is the probability to get the count correct via guessing (here: $p = \frac{1}{9}$) and $k \in [1, 10]$.

**Dataset** We generated an animal counting dataset using Nano Banana. Starting from a white 16:9 image, we used the following prompt, where `number` is in $[1, 9]$ and `animal` is in ['dog', 'elephant', 'cat', 'brown bear', 'horse', 'rabbit', 'raccoon']. We manually evaluated the generated dataset for correctness; the resulting dataset has 54 images (exactly 6 per count).

**Nano Banana**
*Exchange the white space with a realistic photograph of: exactly {number} {animal}, outside, not overlapping, in a natural landscape.*

**Models & prompts** We tested Veo 3 `veo-3.0-generate-preview` and Veo 2 `veo-2.0-generate-preview-001` through the Vertex AI API.

> **Veo**
> *The background changes to white. Then:*
>   - *If there is just a single animal: the animal sits in the middle of the image, looking straight at the camera.*
>   - *If there are multiple animals: all animals line up in a row, with ample white space between them.*

**Sampling**    We generated 10 videos per sample with a fixed prompt.

## B.4    MANIPULATION: IMAGE EDITING

We provide details for the image editing task in Sec. 4.4.

**Evaluation**    We perform a human study with three human raters to evaluate *fidelity* (correct edit) and *precision* (correct edit with no unintended changes like zooming).

**Dataset**    We used a random sample of 30 images from the test set of the Emu-edit dataset [67].

**Models & prompts**    We tested Veo 3 `veo-3.0-generate-preview` and Veo 2 `veo-2.0-generate-preview-001` through the Vertex AI API.

> **Veo**
> *Create a smooth, static animation that slowly {image specific edit direction}. Do not change anything else. No zoom, no pan, no dolly.*

**Sampling**    For each image, we generated two samples and use the first sample for human rating.

## B.5    REASONING: MAZE SOLVING

We provide details for the image editing task in Sec. 4.5.

**Evaluation**    Our evaluation process is tailored to the model type. For Veo, we analyze the generated video frame-by-frame, extracting the path taken by the agent (red circle). We check for any invalid moves, such as jumping over walls, clipping through boundaries, or any alteration of the goal's position. We report the success rate as the fraction of $k$ attempts where the agent successfully reaches the goal (green circle) without any illegal moves.

For Nano Banana, which generates the full path in one edit, we assess whether the drawn path connects the start and end points (allowing for minor discontinuities) and crucially, whether it intersects with any maze walls or goes off the valid path.

For Gemini 2.5 Pro with a maze input as an image (I2T) or as ASCII (T2T), we check whether the series of grid positions represents an uninterrupted path from the start position to the goal.

**Dataset**    For rectangular mazes, we generated 50 random mazes per size using `maze-dataset` [107], but replacing the square start and end with circles and swapping their colors. We also drew 10 irregular mazes by hand and flipped/rotated them to obtain 40 unique samples.

**Models & prompts**    We tested Veo 3 `veo-3.0-generate-preview` and Veo 2 `veo-2.0-generate-preview-001` through the Vertex AI API. We also tested Nano Banana `gemini-2.5-flash-image-preview` and Gemini 2.5 Pro `gemini-2.5-pro` through Google AI Studio.

**Veo**

*Create a 2D animation based on the provided image of a maze. The red circle slides smoothly along the white path, stopping perfectly on the green circle. The red circle never slides or crosses into the black areas of the maze. The camera is a static, top-down view showing the entire maze.*

*Maze:*

- *The maze paths are white, the walls are black.*
- *The red circle moves to the goal position, represented by a green circle.*
- *The red circle slides smoothly along the white path.*
- *The red circle never slides or crosses into the black areas of the maze.*
- *The red circle stops perfectly on the green circle.*

*Scene:*

- *No change in scene composition.*
- *No change in the layout of the maze.*
- *The red circle travels along the white path without speeding up or slowing down.*

*Camera:*

- *Static camera.*
- *No zoom.*
- *No pan.*
- *No glitches, noise, or artifacts.*

**Gemini 2.5 Pro I2T**

SYSTEM
*Think step by step as needed and output in xml format:*
*<think>thinking process</think>*
*<final_answer>final answer</final_answer>*

USER
*The following image shows a maze, represented by colored squares:*

- *Black squares represent walls and cannot be passed through.*
- *White squares are empty and can be passed through.*
- *The red square is the starting point.*
- *The green square is the end point.*

*Please solve the maze by providing a path from the starting point to the end point. The path should be provided as a list of coordinates of each step, where each coordinate is a (row, col) tuple, and row, col are 0-based indices. Consider the origin (0, 0) to be the top-left corner. Overall, the path should be provided in the format of [(row1, col1), (row2, col2), ...].*

*A valid path must:*

- *Start at the starting point (the red square).*
- *End at the end point (the green square).*
- *Avoid the walls (the black squares).*
- *Pass only through empty space (the white squares).*
- *Move one square at a time.*
- *Only move up, down, left, and right, not diagonally.*

*Correct your answer if you spot any errors.*

*Here is the maze image: {image}*

**Nano Banana**
*Mark the correct path from the red to the green circle through the maze in blue.*

**Gemini 2.5 Pro T2T**

SYSTEM

*Think step by step as needed and output in xml format:*
*<think>thinking process</think>*
*<final_answer>final answer</final_answer>*

USER

*The following is an ASCII-representation of a maze:*

- *'#' represents walls which cannot be passed through.*

- *' ' represents empty spaces that can be passed through.*

- *'S' is the starting point.*

- *'E' is the end point.*

*Please solve the maze by providing a path from the starting point to the end point. The path should be provided as a list of coordinates of each step, where each coordinate is a (row, col) tuple, and row, col are 0-based indices. Consider the origin (0, 0) to be the top-left corner. Overall, the path should be provided in the format of [(row1, col1), (row2, col2), ...].*

*A valid path must:*

- *Start at the starting point 'S'.*

- *End at the end point 'E'.*

- *Avoid the walls '#'.*

- *Pass only through empty space ' '.*

- *Move one square at a time.*

- *Only move up, down, left, and right, not diagonally.*

*Correct your answer if you spot any errors.*

*Here is the maze in ASCII format: {maze}*

**Sampling** We generated 10 videos per sample with a fixed prompt. Note that for Gemini 2.5 Pro I2T, we represented the maze as a grid where the red and green positions are marked as squares (not circles) to make the setup grid-like (i.e., a matrix with cells), since this might be easier for a language model.

B.6   **REASONING**: VISUAL SYMMETRY SOLVING

We provide details for the visual symmetry task in Sec. 4.6.

**Evaluation** We prompt Veo with input images containing a $10 \times 16$ grid where a pattern is drawn on the left half. The goal is to complete the pattern on the empty right half so that the final pattern is symmetrical along the central vertical axis.

We compare Veo's best-frame and last-frame solutions with the ground-truth symmetrical grid and compute the number of incorrectly-colored cells. A cell is determined as incorrectly-colored if the average color across pixels in the cell is perceptually distinct from the ground-truth average color in the matching cell. We compute perceptual color differences of the average cell color in the CIELAB color space, with a difference threshold of 15.0. In Fig. 8, we report the percentage of attempts in which the best or last frame solution has zero incorrect cells for $k = 1$.

**Dataset** We created a synthetic grid coloring image dataset to evaluate visual symmetry. We generated 25 samples using common symmetrical symbols, objects and shapes such as english letters (e.g., A, H, M, X), geometric shapes (e.g., square, triangle), symmetrical objects (e.g., wineglass,

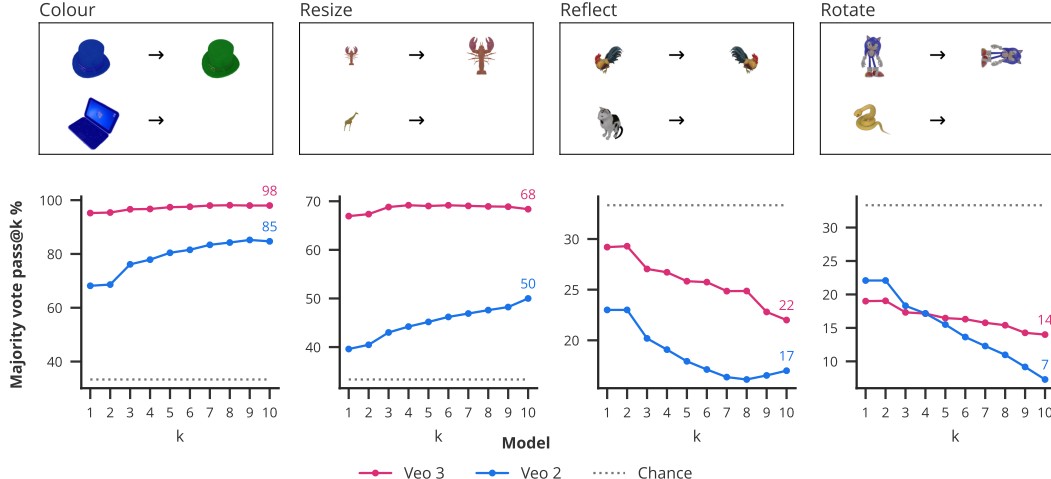

Figure 61: **Visual analogy performance over 10 attempts**. In contrast to other plots in this paper, we here report not the best performance over $k$ attempts, but instead the performance when choosing the *majority vote* from $k$ attempts. As a result, performance is not necessarily monotonic in $k$. In fact, for *reflect* and *rotate*, performance *decreases* with $k$, indicating that both models have systematic, erroneous biases. In the case of Veo 3, the model tends to perform reflections and rotations, but not along the same axis as shown in the image. Veo 2 simply tends to copy the object without applying any transformation.

balloon; together, the *shape* condition). We also generated 25 samples consisting of randomly-colored cells (the *random* condition).

**Models & prompts**  We tested Veo 3 `veo-3.0-generate-preview` and Veo 2 `veo-2.0-generate-preview-001` through the Vertex AI API. We also tested Nano Banana `gemini-2.5-flash-image-preview` through Google AI Studio.

> **Veo**
> *Instantly reflect this pattern along the central, vertical axis while keeping the existing colored pattern without modification. Static camera perspective, no zoom or pan.*

**Sampling**  We generated 10 videos per sample with a fixed prompt.

### B.7 REASONING: VISUAL ANALOGY COMPLETION

We provide details for the visual analogy task in Sec. 4.7.

**Evaluation**  We prompt Veo to solve visual analogies with an input image showing a reference object pair and a test object. The object images are sourced from the Kid-inspired Visual Analogies benchmark [KiVA, 70]. Consistent with the multi-choice format in the KiVA benchmark, we evaluated Veo's generation by cropping out the generated target object in the lower-right region of the last frame and compare Veo's generated object with three candidate object choices using an autorater (see details below).

In Fig. 9, we report the pass@1 accuracy across different conditions for both Veo 2 and Veo 3 for $k = 1$. Fig. 61 shows performance for $k = 10$.

**Dataset**  We used the test trials and choice images from the KiVA benchmark [70].

**Models & prompts**  We tested Veo 3 `veo-3.0-generate-preview` and Veo 2 `veo-2.0-generate-preview-001` through the Vertex AI API.

We used Gemini 2.5 Pro `gemini-2.5-pro` through Google AI Studio to identify which image choice Veo's generation is most similar with. To enhance the autorater's image comparison accuracy for this task, Gemini is prompted with privileged information about the values in the dataset conditions (see below for the full autorater prompt). If no object is visible in the lower-right region of Veo's generated last frame or if the generated object is of a different object type, we randomly sampled one of three choices as Veo's choice. In pilot experiments, we found that the Gemini-assisted autorater's ratings achieve above 88% agreement with expert human ratings by the authors on 25 samples within each conditions.

Note that in the prompt, words in { } are updated based on the test condition of the current generation (one of *color*, *resize*, *reflect*, and *rotate*) to provide more information of the feature name and values to direct the image comparison. Image choice orders are shuffled for each prompt.

---

**Veo**

*Create a smooth animation to generate the missing object in the lower right region and solve the visual analogy. The original three objects must remain still. Static shot, no zoom no pan no dolly.*

---

**Gemini 2.5 Pro autorater**

SYSTEM

*You are an expert visual judge. You will be presented with a "target image" and three "choice images" labeled A, B, and C. Your goal is to identify the choice image that is most visually similar to the target image.*

*Follow these steps:*

  1. *Analyze each provided image and describe the objects shown. Focus on the object {color}. That is, if the objects appear {green}, {blue}, or {red}.*

  2. *Determine if the primary object in the target image is of the same general category or type as the objects in the choice images. For example, if the target image shows a dog, and the choices show a cat, the object types are considered different. If no object is visible in the target image, the object type is considered to be mismatched.*

  3. *If the object type matches between the target image and the choice images, identify the choice that is most visually similar to the target image in terms of the object {color}.*

*Provide a brief justification for your choice, explaining why it is the best match and why the others are less suitable. Conclude your response with the final answer on a new line in the format:*
*"Final Answer: [answer]"*
*where "answer" is one of ("A", "B", "C", or "different object type"). Do not use markdown format for the final answer line.*

USER
*Please evaluate the following images.*
*— TARGET IMAGE —*
*{target object image}*
*— CHOICE IMAGES —*
*CHOICE A: {image choice} CHOICE B: {image choice} CHOICE C: {image choice}*
*Which choice image is most similar to the target image?*

---

**Sampling** We generated 10 videos per sample with a fixed prompt.

## C  Prompting best practices

Table 2: **Prompt sensitivity study on the visual symmetry task**. We report best frame pass@1 % and the average number of incorrectly-colored cells across 25 samples on each split (shape/random).

| | | Pass@1 | | Avg incorrect cells | |
|---|---|---|---|---|---|
| No. | Prompt | Shape | Random | Shape | Random |
| 1 | Instantly reflect this pattern along the central, vertical axis while keeping the existing colored pattern without modification. | 48 | 68 | 4.16 | 7.00 |
| 2 | Instantly reflect this pattern along the central, vertical axis while keeping the existing colored pattern without modification. Static camera perspective, no zoom or pan. | 42 | 65 | 5.00 | 3.52 |
| 3 | Instantly reflect this pattern along the central, vertical axis while keeping the existing colored pattern without modification. The result needs to be mirror-symmetrical along the vertical axis. Static camera perspective, no zoom or pan. | 36 | 52 | 6.28 | 9.04 |
| 4 | One by one, cells in the right half of the grid are filled in to complete the pattern. The pattern is mirror-symmetrical along the central vertical line. Static shot; no zoom, no pan, no dolly. | 32 | 12 | 10.76 | 14.08 |
| 5 | Reflect this pattern along the central, vertical axis. | 28 | 40 | 9.76 | 4.52 |
| 6 | An animation showing the left half of the grid being mirrored onto the right half to create a symmetrical pattern. Static shot; no zoom, no pan, no dolly. | 24 | 12 | 10.96 | 16.32 |
| 7 | You're a master symmetry solver. Your task is to fill the cells on the right side of the grid to mirror the pattern on the left, such that it's symmetrical along the vertical axis. | 24 | 8 | 9.20 | 17.72 |
| 8 | Fill color in the appropriate cells on the right side of the grid to complete the pattern. The final image should be symmetrical along the central vertical line. Static shot, no zoom no pan no dolly. | 13 | 9 | 10.30 | 14.74 |
| 9 | Create a static, smooth, realistic animation completing the pattern in the image by filling the grid on the right hand side. Do not change anything else. No zoom, no pan. | 12 | 4 | 14.88 | 21.00 |
| 10 | A timelapse of a professional pixel artist drawing a symmetrical pattern onto a white canvas. Static shot; no zoom, no pan, no dolly. | 8 | 20 | 14.20 | 12.64 |

The results in Secs. 3 and 4 are best-effort estimates of Veo's performance using carefully chosen prompts. Generally, performance varies greatly with the exact task description provided in the prompt, as illustrated by a prompt sensitivity study on the visual symmetry task in Table 2. Here are best practices from this sensitivity analysis and our other experiments:

- **Remove ambiguity**. Tasks can be solved in a variety of ways, and natural language descriptions tend to leave a lot of room for interpretation. The goal should be formulated clearly, e.g., saying "symmetrical along the central, vertical axis", rather than just "symmetrical".
- **Specify what shouldn't change**. Veo has a tendency to change any part of the input to create interesting, dynamic scenes. Including not only a positive task description, but also specifying what *not* to change can help mitigate this, e.g., "keep the existing colored pattern without modification".
- **Providing an outlet**. As mentioned above, Veo has a strong prior to keep things moving. Providing a "motion outlet" in the form of, e.g., a spinning ball can help keep the rest of the scene static.
- **Let the model decide when its done**. The motion prior also means that Veo often keeps modifying the scene, even after solving the task. Providing a visual indicator, e.g., "add a

glowing red dot once the goal is reached" allows for easy extraction of the solution from the generated video.

- **Scene and camera controls**. Phrases like "static camera, no zoom, no pan, no dolly" can help keeping the scene static, e.g., for image-to-image tasks.

- **Speed control**. Some tasks like maze solving benefit from being solved step-by-step. For other tasks, especially image-to-image tasks, specifying instant changes can help avoid artifacts.

- **Realism**. Veo was trained to generate plausible, realistic-looking videos. Translating an abstract task into a realistic setting (including, but not limited to editing the original image to depict realistic, 3D scenes rather than abstract shapes) can improve generation results. A similar effect was observed in [108], and we expect *visual prompt engineering* to emerge as a powerful tool for video models.

## D   VEO INFERENCE DETAILS

NEW

For both qualitative and quantitative tasks, we generate videos via the Vertex AI API. We do not manually specify a seed, making generation non-deterministic and subject to Vertex's default sampling behavior. All videos are generated without audio and with a default length of eight seconds.

At the time of writing, generation costs 0.20 USD/s for Veo 3 and 0.50 USD/s for Veo 2.

Video generation took 18.2s on average per video; this includes not only the time to generate the video but also the potential waiting time until resources become available and the computation is scheduled, as well as the time it takes to transfer the input from our local machine to the datacenter and the output video back to the local machine. In principle, all of these steps can be parallelized (e.g., when running a larger dataset), subject to network bandwidth and compute availability.

## E  FAILURE CASES

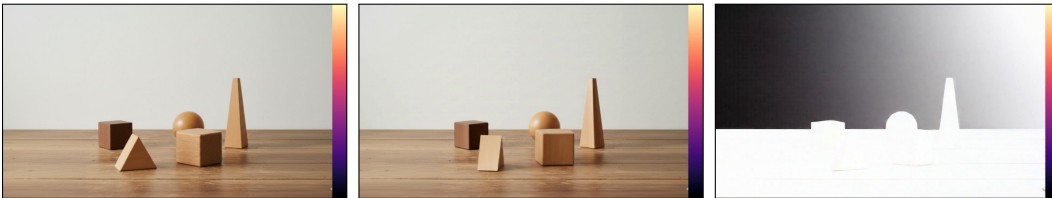

Figure 62: **Monocular depth estimation**. Prompt: *"The image transitions to a depth-map of the scene: Darker colors represent pixels further from the camera, lighter colors represent pixels closer to the camera. The exact color map to use is provided on the right side of the image. Static scene, no pan, no zoom, no dolly."* Failure: Veo 3 seems generally unable to color pixels by depth beyond a binary foreground/background mapping and specifically struggles with using a provided color map.

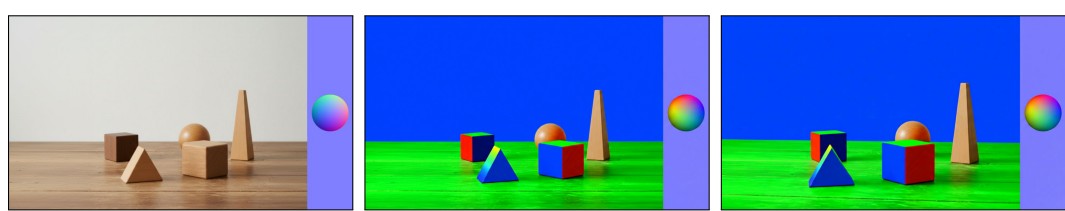

Figure 63: **Monocular surface normal estimation**. Prompt: *"The image transitions to a surface-normal map of the scene: the red/green/blue color channel specify the direction of the surface-normal at each point, as illustrated on the right side of the image on a sphere. Static scene, no pan, no zoom, no dolly."* Failure: While Veo 3 shows some promise in coloring surfaces according to their orientation (e.g., the cube in the front), coloration is inconsistent (compare the two cubes) and doesn't correctly interpolate colors (e.g., for the slope on the triangle).

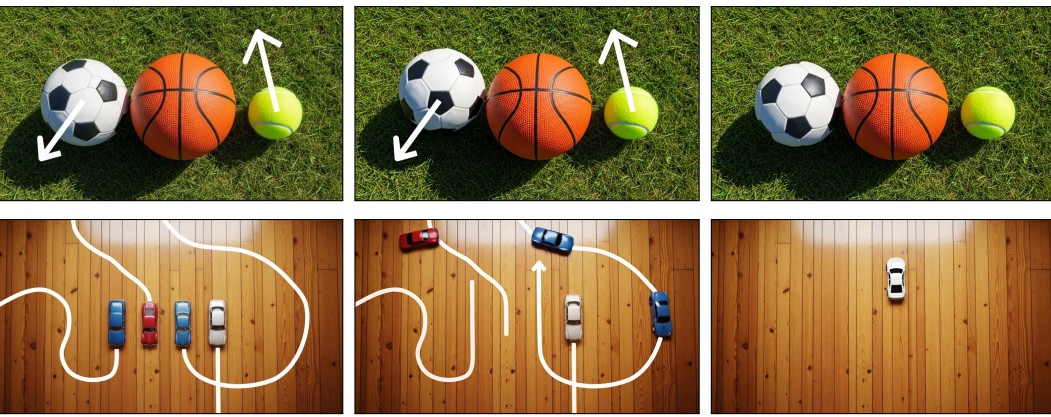

Figure 64: **Force & motion prompting**, inspired by [109, 110]. **Force prompting** (top). Prompt: *"The balls move in the direction indicated by the arrows. Balls without an arrow don't move. Static scene, no pan, no zoom, no dolly."* **Motion trajectory prompting** (bottom). Prompt: *"Each car drives out of the frame following the indicated trajectory. Static camera, no zoom, no pan, no dolly."* Failure: Veo 3 seems unable to follow force/motion annotations with any consistency. Providing annotations for the first frame and letting the model remove them before generating the scene in motion does not work, either.

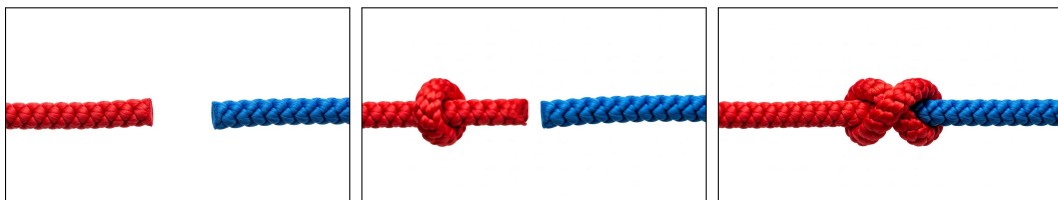

Figure 65: **Tying the knot**. Prompt: *"A knot is tied connecting these two rope ends."* Failure: physics violation, impossible rope movement.

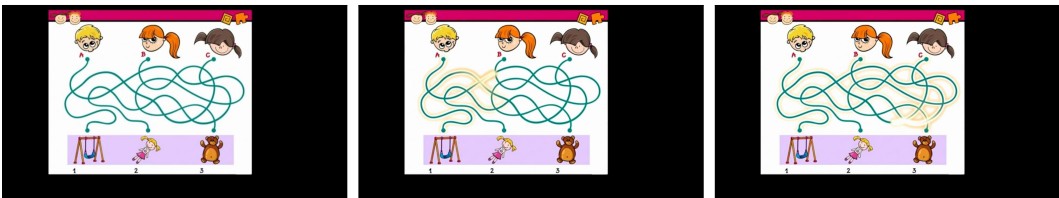

Figure 66: **Connect the path puzzle**. Prompt: *"The path connecting the boy to the object starts glowing slowly. Nothing else changes. No zoom, no pan, no dolly."* Failure: hallucinations, lighting up of all paths.



Figure 67: **Five letter word search**. Prompt: *"Generate a static video animation using the provided letter grid. The task is to highlight the only 5-letter English word CHEAT, which may be oriented in any direction (horizontally, vertically, or diagonally). The animation should consist of a semi-transparent red rectangle with rounded corners smoothly fading into view, perfectly encapsulating the five letters of the word. The rectangle should have a subtle, soft glow. Do not change anything else in the image. The camera must remain locked in place with no movement. No zoom, no pan, no dolly."* Failure: does not recognize words; highlights individual letters randomly.

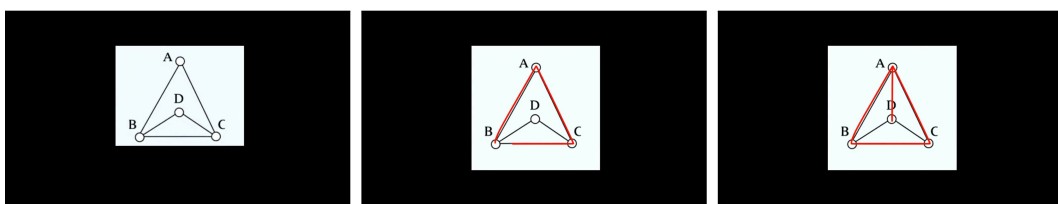

Figure 68: **Eulerian path**. Prompt: *"Create a smooth animation where a red pen traces all existing edges in a continuous path without lifting the pen. All edges need to be traced. Do not visit any edge twice and do not lift the pen. No zoom, no pan."* Failure: does not trace the edges exactly, traces non-existent edges.

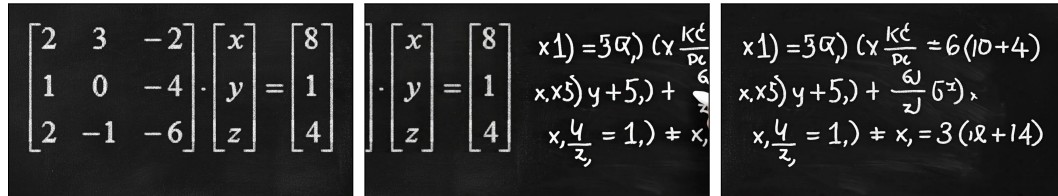

Figure 69: **Solving system of linear equations**. Prompt: *"A hand appears and solves the set of linear equations. It replaces the x, y, z matrix with their correct values that solves the equation. Do not change anything else."* Failure: hallucinations with text on the blackboard.

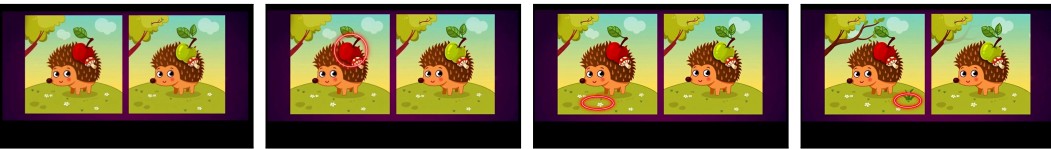

Figure 70: **Spot the difference**. Prompt: *"There are two images. The left image is different from the right image in 5 spots. Create a static, realistic, smooth animation where a cursor appears and points at each place where the left image is different from the right image. The cursor points one by one and only on the left image. Do not change anything in the right image. No pan. No zoom. No movement. Keep the image static."* Failure: does not identify all the differences. Hallucinates differences.

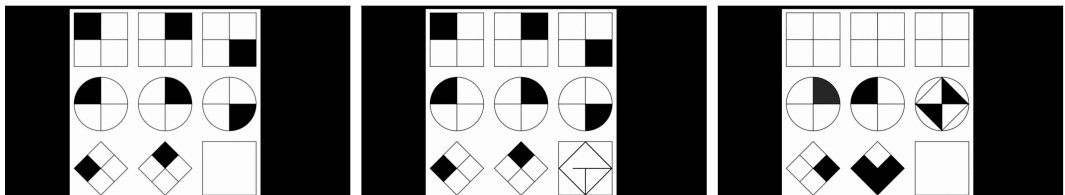

Figure 71: **Visual IQ test**. Prompt: *"Create a static, smooth, animation that solves the puzzle in the given image. The correct pattern should appear at the bottom right to solve the puzzle. Do not change anything else in the picture. No zoom, no pan, no dolly"* Failure: incorrect figure pattern.

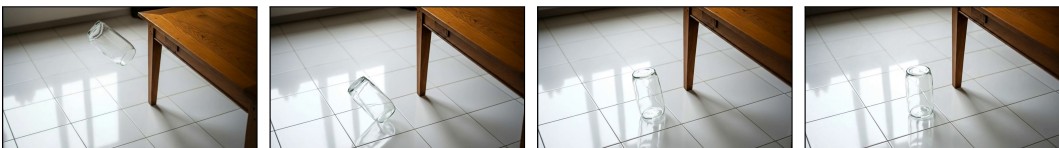

Figure 72: **Glass falling**. Prompt: *"The object falls. Static camera, no pan, no zoom, no dolly."* Failure: physics violation, glass does not break, and orients itself to be vertical after landing on the floor.

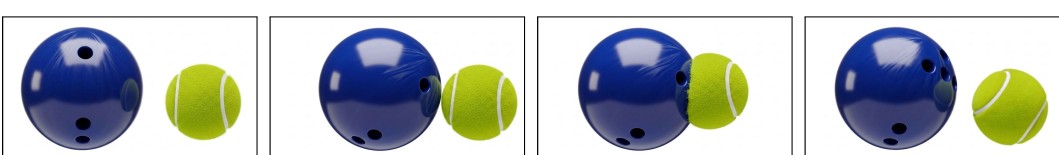

Figure 73: **Collisions**. Prompt: *"The two objects collide in slow motion. Static camera, no pan, no zoom, no dolly."* Failure: not physically plausible, the objects pause at the moment of impact and then are pushed together by an invisible force.

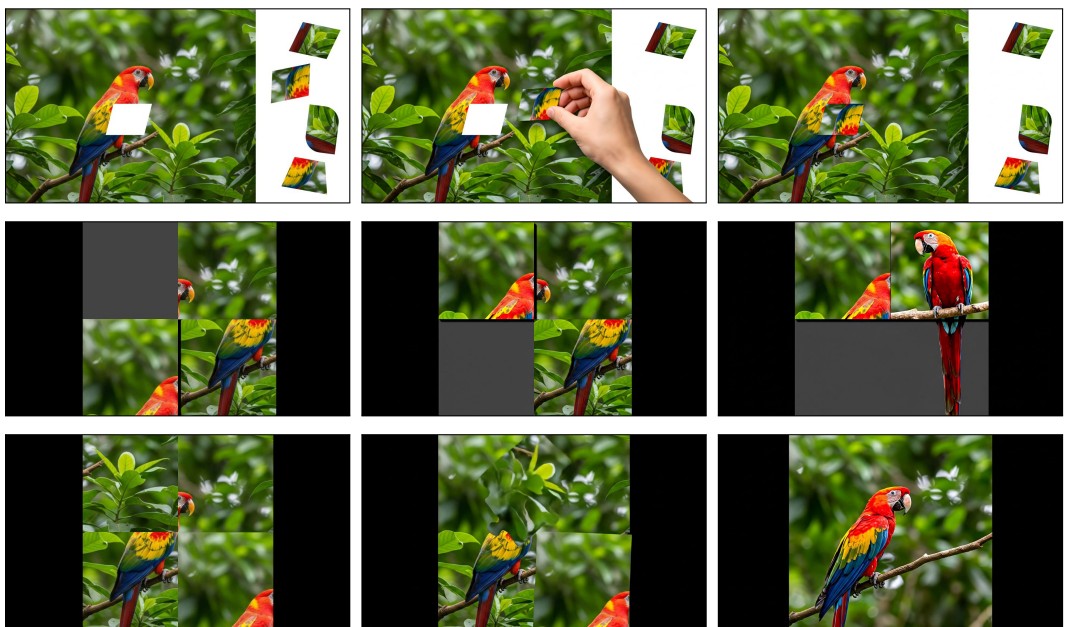

Figure 74: **Tiling puzzles**. **Jigsaw puzzle** (top). Prompt: *"A hand takes the fitting puzzle piece from the right, rotates it to be in the correct orientation, then puts it into the hole, completing the puzzle. Static scene, no pan, no zoom, no dolly."* Failure: wrong piece orientation. **Sliding puzzle** (middle). Prompt: *"Slide the pieces of this sliding puzzle around one-at-a-time until all edges align."* Failure: doesn't maintain piece integrity while sliding, hallucinates new pieces. **Scrambled puzzle** (bottom). Prompt: *"Unscramble this image."* Failure: image details are inconsistent with original pieces.

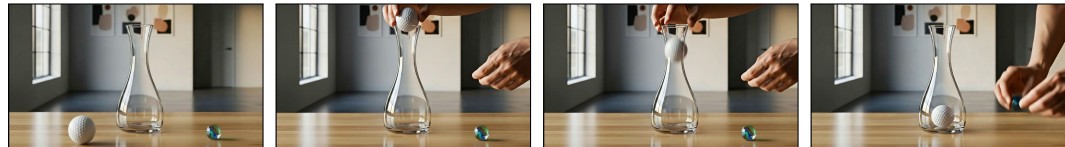

Figure 75: **Bottleneck**. Prompt: *"A person tries to put the golf ball in the vase. Static camera, no pan, no zoom, no dolly."* Failure: not physically plausible, golf ball is too large to pass through the bottleneck of the vase.

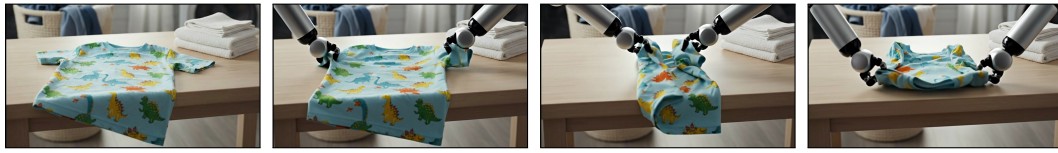

Figure 76: **Laundry folding**. Prompt: *"Generate a video of two metal robotic arms properly folding the t-shirt on the table.* Failure: physics violation, implausible folding movements.

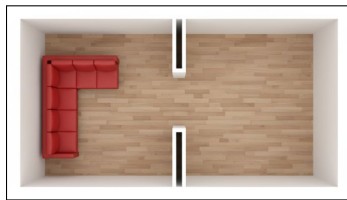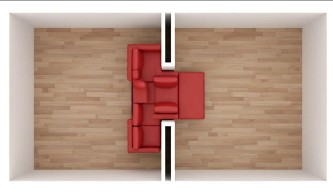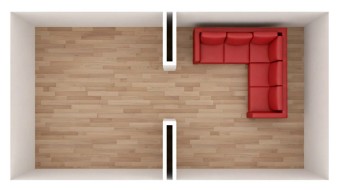

Figure 77: **Motion planning**; inspired by the piano mover's problem. Prompt: *"The red couch slides from the left room over into the right room, skillfully maneuvering to fit through the doorways without bumping into the walls. The walls are fixed: they don't shift or disappear, and no new walls are introduced. Static camera, no pan, no zoom, no dolly."* Failure: violating rigid-body integrity, not keeping to permissible transformations (rotation, translation).

## F    LLM USE

Gemini 2.5 Flash and Gemini 2.5 Pro [2] were used for brainstorming task ideas, suggesting related work that we might have otherwise missed, coding support, and to polish human writing.

## G    IMAGE SOURCES

Where not stated in the figure caption, images were obtained as follows.

- Figs. 10 to 15, 32 to 38 and 74: The original macaw image was generated with Gemini and, depending on the figure, subsequently modified by the authors (e.g., conversion to grayscale, adding noise, adding the monkey with Nano Banana).
- Fig. 16: The input image was obtained from here (Apache 2.0 license) based on the LOLv2 dataset [93] and randomly selected. The image was slightly cropped to fit a 16:9 aspect ratio.
- Figs. 17, 21 to 24, 26 to 29, 31, 39, 41, 42, 46, 47, 52, 54, 65, 69, 72, 73 and 75 to 77: generated with Gemini.
- Fig. 25: The input image was obtained from here (CC0 license).
- Fig. 30: hand drawn by us, inspired by Fig. 1 of the Omniglot paper [56].
- Fig. 40: sample from Objaverse [98]
- Figs. 48 to 51, 53, 55 and 57: created by us.
- Figs. 56, 66, 67 and 70: original image from Reddit.
- Fig. 59: hand drawn by us, inspired by ARC-AGI [99].
- Fig. 60: sample from BIPEDv2 [64, 65].
- Figs. 62 to 64: generated with Gemini, then annotated by us.
- Figs. 68 and 71: hand drawn by us. Inspired by original images from Reddit.
- Figs. 44 and 45: The robot hands are extracted from a frame in this video and were subsequently adapted with Nano Banana. The hands holding Baoding balls were obtained from here.

