# OpenReview forum: "Video models are zero-shot learners and reasoners"
_ICLR.cc/2026/Conference — Submitted to ICLR 2026_

### Official Review · Reviewer_qvjV · 2025-10-29

**Soundness:** 2
**Presentation:** 3
**Contribution:** 3
**Rating:** 4
**Confidence:** 3

**Summary:**

The paper investigates the zero-shot performance of Veo 2 and Veo 3 using a variety of different tasks. The evaluations are grouped into four categories, namely perception, modeling, manipulation, and reasoning. The results indicate emergent capabilities in these tasks, even though they were presumably not trained on them. The success in solving these tasks is compared to the success of Large Language Models (LLMs), and it is argued that, just like LLMs, general vision foundation models will replace specialized models.

**Strengths:**

1. Vision foundation models are a highly relevant and important topic, and investigating their abilities and shortcomings is of great interest to the scientific community.
2. The proposed tasks across the four categories (Perception, Modeling, Manipulation, Reasoning) are chosen well and help categorize how the model performs in different situations.
3. Both qualitative and quantitative evaluations are conducted, and the qualitative evaluation is extensive and covers a wide variety of different tasks.
4. A project page is included, making it possible for the reader to look at the videos directly, which is not possible in a PDF.

**Weaknesses:**

My main concern is that the paper contains several claims that are not adequately supported or discussed. The supporting discussion and evidence should be expanded, or the claims moderated accordingly. Specifically, I am referring to the following statements:

1. "Just like LLMs replaced task-specific NLP models". That statement seems debatable. LLMs are very powerful, but the claim that they fully replaced task-specific models requires further discussion or evidence. Saying that task-specific NLP models have been entirely replaced is incorrect [1,2].
2. "Once they become sufficiently cheap and reliable". Is it reasonable to think foundation models could be as cheap and reliable as specialized models that focus, e.g., on edge detection or object grounding? Or is the reasoning behind this claim different? For example, one might argue that such models might never be cheaper in direct comparison, but cheap enough that the small cost is insignificant. In either case, this requires further discussion.
3. "If NLP is a guide, the same trend will play out in vision". A discussion on the similarities and differences between NLP and vision would be a helpful contribution to this paper. What are the similarities between NLP and vision that would indicate that the same pattern will play out? On the other hand, what are the differences and challenges faced by vision that might be challenging for vision foundation models?
4. The chain-of-frame concept is interesting but could be discussed in more detail. It seems a plausible explanation for visual reasoning performance; however, its effectiveness should be further validated, e.g., through comparison with models that generate the final image directly instead of through frame-by-frame generation. The comparison in Section 4.6 to Nano Banana is a good indicator, but more discussion and different tasks would be beneficial.

[1] Pecher, Branislav, Ivan Srba, and Maria Bielikova. "Comparing specialised small and general large language models on text classification: 100 labelled samples to achieve break-even performance." *arXiv preprint arXiv:2402.12819* (2024).

[2] Xu, Zongzhe, et al. "Specialized Foundation Models Struggle to Beat Supervised Baselines." *The Thirteenth International Conference on Learning Representations* (2025).

**Questions:**

1. Do I understand correctly that the authors manually evaluated most of the model's outputs? If so, was it always straightforward to determine whether the model’s output should be considered a success or a failure? I would expect many cases where the model’s response is partially correct but not entirely accurate. If such ambiguities occurred, how were they handled?
2. In Figure 56, the example does not appear to be solved correctly (glass 5 should fill up, but then the pipe is blocked in the third image). Is this on purpose, and was this counted as a correct example?

---

> ### Author Response · Authors · 2025-11-20
> **Response to reviewer qvjV (1/2)**
>
> Dear reviewer qvjV,
>
> Thank you for your thoughtful review of our work, and for noting it as addressing a **highly relevant and important topic**. We are pleased to see that you find our experiments to be **extensive** and **well-chosen**. We hope that the additional experiments and explanations/revisions of statements in the article based on your comments further strengthen our contribution.
>
> > My main concern is that the paper contains *several claims that are not adequately supported or discussed. The supporting discussion and evidence should be expanded, or the claims moderated accordingly*. Specifically, I am referring to the following statements:
>
> > 1\. "Just like LLMs replaced task-specific NLP models". That statement seems debatable. LLMs are very powerful, but the claim that they fully replaced task-specific models requires further discussion or evidence. Saying that task-specific NLP models have been entirely replaced is incorrect \[1,2\].
>
> That’s a great point, thanks for flagging\! We fully agree, and stating that all task-specific NLP models have been replaced was not at all our intention. We intended to say that *many* task-specific models have been replaced by LLMs, not *all* of them. Our statements in the paper were imprecise & **we’ve now corrected Takeaway 2** to state “Just like LLMs replaced many task-specific NLP models” and also adapted Takeaway 1 accordingly by adding “for many tasks”.
>
> > 2\. "Once they become sufficiently cheap and reliable". *Is it reasonable to think foundation models could be as cheap and reliable as specialized models* that focus, e.g., on edge detection or object grounding? Or is the reasoning behind this claim different? For example, one might argue that such models might never be cheaper in direct comparison, but cheap enough that the small cost is insignificant. In either case, this requires further discussion.
>
> Another excellent point where our discussion needs more nuance. In terms of cost: Given that foundation models tend to be “big”, we believe it’s unlikely that they’ll ever be as cheap as a specialized model that’s been trained to solve one task, and one task only. However, as they become better, cheaper & more reliable over time, foundation models often reach a point where their inference cost is justified by the performance improvement over individual, task-specific models. For example, in NLP, today’s best coding models (see e.g. [https://www.swebench.com/](https://www.swebench.com/) or [https://web.lmarena.ai/leaderboard](https://web.lmarena.ai/leaderboard)) aren’t coding-only models but general-purpose LLMs instead. The reason for this is that there are synergies between related tasks: e.g., training a model on math \+ coding might be better than training one model on math and another one on coding. Similarly, in vision, many tasks have deep relationships with each other (as shown e.g. by [https://arxiv.org/pdf/1804.08328](https://arxiv.org/pdf/1804.08328)). Therefore, a unified visual foundation model for many different tasks might develop such a good understanding of the visual world that its outstanding performance across tasks is very clearly worth the computational cost. Of course, we’re not there yet, and your references \[1\] and \[2\] are important reminders that we shouldn’t underestimate specialized models either, but the fact that general-purpose LLMs have become standard tools for software development (a specialized task) is, in our eyes, a strong indication that general-purpose models could see widespread adoption in vision, too, once their performance justifies the cost. We hope this explains our reasoning better, and we now include this discussion in Section 5 (end of paragraph “Video generation is expensive …”).

---

> ### Author Response · Authors · 2025-11-20
> **Response to reviewer qvjV (2/2)**
>
> > 3\. "If NLP is a guide, the same trend will play out in vision". *A discussion on the similarities and differences between NLP and vision would be a helpful contribution to this paper*. What are the similarities between NLP and vision that would indicate that the same pattern will play out? On the other hand, what are the differences and challenges faced by vision that might be challenging for vision foundation models?
>
> We appreciate the suggestion and have added the following discussion to the paper (section Discussion, subsection “Similarities and differences with NLP”): There are strong reasons to believe that progress in vision will mirror the rapid advancements seen in NLP. Firstly, both language and vision models are developed on the same hardware, meaning that vision models benefit from the same hardware scaling laws (e.g., Moore's law) that have driven down the cost of training and inference for LLMs. Secondly, many of the pivotal algorithmic breakthroughs, such as the Transformer architecture, efficient attention mechanisms, and skip connections, are modality-agnostic and have been successfully adapted for both domains. For these reasons, just as inference costs for language models are rapidly decreasing, we can expect a similar trend for vision models, which will be a key enabler of their widespread adoption.
>
> However, there are also crucial differences. A significant hurdle for vision is the cost of training. Video data is far more information-dense and computationally expensive to process than text, which could lead to a slower development and adoption cycle for large-scale vision models compared to their NLP counterparts. Conversely, the issue of data availability presents a long-term advantage for vision. It is widely believed that we are approaching the limits of available high-quality text data for LLM training. In contrast, the amount of video data being generated globally continues to grow at an exponential rate, far outstripping what is currently used for model training. This suggests that while training costs are a near-term challenge, the vast and growing reservoir of video data may ultimately fuel the development of even more powerful and capable foundation models for vision in the long run.
>
> > 4\. *The chain-of-frame concept is interesting but could be discussed in more detail / its effectiveness should be further validated, e.g., through comparison with models that generate the final image directly instead of through frame-by-frame generation*
>
> We thank the reviewer for their feedback on the chain-of-frame (CoF) concept and more validation with models that generate only a final image. To address these we have taken the following steps:
>
> First, to address your suggestion of expanded validation, we have expanded our comparison between Veo and Nano Banana (NB) as an image generation model to encompass all three visual reasoning tasks explored in our article: maze solving, symmetry solving, and visual analogy solving (updated Figure 9). On visual analogies, the results show that Veo and NB are at chance for Rotation and Reflection, while NB performs better on Resize whereas Veo 3 is better in Colour; thus this is an example of a task where chain-of-frames doesn’t seem to provide a benefit, while for irregular mazes and symmetry solving there is a clear advantage of the video model. Second, we have added a sentence to the discussion section to more explicitly frame the comparisons with Nano Banana (now across four different tasks: edge detection, segmentation, maze solving, and visual analogies). Together, these results allow for a more accurate assessment of which tasks benefit from solving them with a video model (producing a chain of frames) vs. which tasks can also be solved with a generative image model.
>
> > 5\. *Fig.56, the example does not appear to be solved correctly*.
>
> The pipe connecting to container 5 is blocked at the outlet (as can be seen in the input image when zooming in) and therefore no water should flow into container 5\. The model thus indeed fills the correct glasses in the right order. Since this detail is easily missed, we’ve now updated the figure caption to explain the solution.

---

### Official Review · Reviewer_k3zu · 2025-10-30

**Soundness:** 3
**Presentation:** 3
**Contribution:** 3
**Rating:** 6
**Confidence:** 3

**Summary:**

This paper demonstrates through extensive experimentation that modern generative video models (specifically Veo 3) possess remarkable zero-shot capabilities across a wide spectrum of vision tasks—from perception to reasoning—suggesting their potential to evolve into general-purpose vision foundation models, much like LLMs in NLP. The key evidence is the substantial performance leap from Veo 2 to Veo 3, indicating rapid progress in this domain.

**Strengths:**

1.	The paper demonstrates remarkable originality by systematically exploring and validating the provocative thesis that video generation models are emergent, zero-shot generalist vision models, a novel conceptual contribution to the field.
2.	The quality of the empirical evidence is exceptionally high, supported by a massive-scale evaluation across 62 qualitative and 7 quantitative tasks, which makes the core argument both convincing and a significant benchmark for future work.
3.	The work possesses substantial significance as it provides compelling evidence for an impending paradigm shift in computer vision, suggesting a move away from specialized models towards unified foundation models, similar to the transformation witnessed in NLP.

**Weaknesses:**

1.	While the paper compellingly demonstrates the emergent zero-shot capabilities of video models, it remains a capability showcase and lacks analysis of the underlying mechanisms. A discussion on whether these abilities stem from scale, data, or the video generation objective itself would provide crucial academic depth.
2.	The generational improvement from Veo 2 to Veo 3 is clear, but the claim of being a general-purpose vision foundation model requires benchmarking against other state-of-the-art models (e.g., Sora, Dreamix) to properly establish its standing in the field.
3.	Using best-frame and pass@k (k ≤ 10) inflates results via repeated sampling and selection. This does not reflect realistic usage. The paper should report compute budgets, sampling settings, and variance, and include pass@k vs k and accuracy vs cost curves to support fair comparison.
4.	While the paper demonstrates Veo 3's performance on various tasks, it does not analyze how the model makes its decisions. Understanding the reasoning process in visual reasoning tasks is crucial. Did the authors explore how Veo 3 performs reasoning, especially on complex tasks? Without explainability, the work lacks depth, making it difficult to assess the model’s true potential.
5.	The compelling argument for video models as general-purpose vision foundations is critically undermined by the lack of discussion on practical viability. The claim of potentially replacing efficient, specialized models hinges not only on capability but also on computational efficiency. We strongly recommend the authors address this by including a discussion on the inference cost and latency observed in this study (e.g., via API pricing and generation times), and to explore pathways for future efficiency improvements (e.g., via model distillation or quantization). Without this, the "replacement" thesis lacks credibility and practical guidance for the community.

**Questions:**

review the weaknesses

---

> ### Author Response · Authors · 2025-11-20
> **Response to reviewer k3zu (1/2)**
>
> Dear reviewer k3zu,
>
> Thank you very much for your thoughtful feedback. We appreciate you highlighting the **remarkable originality** and **exceptionally high quality of empirical evidence** of our study. We trust that the additional details and discussion based on your comments further improve the **substantial significance** of our work for the community.
>
> > 1\. While the paper compellingly demonstrates the emergent zero-shot capabilities of video models, it remains a capability showcase and *lacks analysis of the underlying mechanisms*. *A discussion on whether these abilities stem from scale, data, or the video generation objective itself would provide crucial academic depth*.
>
> Thank you for raising this point, we share your interest in better understanding the origins and mechanisms of these capabilities. **We have added a discussion in Sec. 5** (paragraphs “Will other video models show similar behavior?” and “Where does Veo’s performance come from?”). In a nutshell, we believe three related/follow-up works published after the submission of our work show that emergent behavior isn’t limited to Veo, and therefore not specific to its training recipe or data mixture. The role of model and data scale as well as training compute can only be properly studied via scaling laws, and we hope our work provides a good selection of possible tasks to study in that context.
>
> > 2\. The generational improvement from Veo 2 to Veo 3 is clear, but *the claim of being a general-purpose vision foundation model requires benchmarking against other state-of-the-art models* (e.g., Sora, Dreamix) to properly establish its standing in the field.
>
> Our paper’s primary argument, as reflected in our title, is that video models as a class are on a trajectory to become general-purpose vision foundation models, much like LLMs have for language. We do not claim that Veo 3 is the only or the definitive foundation model for vision. Instead, as we state in Section 2 in the "Why Veo?" subsection, we chose Veo as a representative example to demonstrate this emergent phenomenon because of its publicly available API and its strong performance on leaderboards, which makes it a suitable subject for a case study of a state-of-the-art video mode
>
> Regarding the definition of a "foundation model, we follow the authoritative work “On the Opportunities and Risks of Foundation Models” ([https://arxiv.org/abs/2108.07258/](https://arxiv.org/abs/2108.07258/)), which defines a foundation model as any model “that is trained on broad data (generally using self-supervision at scale)” and which can be adapted (e.g. fine-tuned, prompted, …) “to a wide range of downstream tasks”. Based on our comprehensive results across several downstream tasks (62 qualitative and 7 quantitative), Veo 3 clearly fits this definition, independent of other models that may be or may become visual foundation models as well. **We now explicitly clarify this definition in Sec. 2** (paragraph “approach and motivation”) to explain what we mean by “vision foundation model”. We hope our analysis will spark a broader discussion about the general visual capabilities of video models, beyond content creation. While our article does not propose a model or make comparative statements (e.g., “model X is better than model Y”), we welcome follow-up investigations and comparative benchmarks across different models.

---

> ### Author Response · Authors · 2025-11-20
> **Response to reviewer k3zu (2/2)**
>
> > 3\. Using best-frame and pass@k (k ≤ 10\) inflates results via repeated sampling and selection. This does not reflect realistic usage. *The paper should report compute budgets, sampling settings, and variance, and include pass@k vs k and accuracy vs cost curves to support fair comparison*.
>
> The reviewer raises some fair points, but we believe there might also be a small misunderstanding here:
> Wherever performance @k is reported, we already show performance plots with $k \in \[1, 10\]$ on the x-axis. In all other cases, including the qualitative results in Sec. 3 and image editing performance in Fig. 6, we report results for $k=1$. If our writing can be improved to make this clearer, kindly let us know, we’re always happy to receive suggestions for improvement.
>
> We report performance@k to study the reliability with which models solve a given task, as is standard in, e.g., the LLM post-training literature. Performance@k should always be interpreted as an estimation of a current performance ceiling, not the average expected performance. As this is an analytical tool, it does not reflect practical usage: We agree that video models today are too compute-intensive to make performance@k feasible. Yet, we believe costs will fall drastically over the next few years, enabling repeated sampling similar to what is common practice with LLMs. In this case, simple ensembling methods, e.g., self-consistency (as used in the LLM literature) / majority vote could bring the average performance closer to our reported performance ceiling.
>
> Similarly, where we evaluate best-frame performance, we also show last-frame performance. For all qualitative examples, performance is evaluated on task-specific criteria, as outlined in the newly added App. A.5. Here as well, best-frame performance should be read as a performance ceiling. We include the comparison between best- and last-frame performance as it highlights an important shortcoming of current video models: trained for aesthetics, models exhibit a strong prior to “keep things moving” and do not necessarily stop or “freeze” when a solution is found before the end of the video. This penalizes performance on image tasks like edge detection and therefore risks underestimating a model’s inherent performance “opportunity” (that could be achieved by e.g. fine-tuning the model to avoid moving the camera).
>
> In response to your question about cost, **App. D now lists the cost for Veo inference**. We seek to demonstrate the potential of video models for visual intelligence tasks. Our understanding is that many tasks in AI are initially solved with a more expensive approach, and once sufficiently many people become interested in the task, there are often ways to improve the efficiency without sacrificing performance (e.g. ChatGPT \-\> DeepSeek, or big foundation models \-\> smaller distilled models with similar performance). Our work clearly just focuses on the first step \- showing that it’s possible to solve challenging tasks with a video model.
>
> > 4\. While the paper demonstrates Veo 3's performance on various tasks, *it does not analyze how the model makes its decisions*. Understanding the reasoning process in visual reasoning tasks is crucial. Did the authors explore how Veo 3 performs reasoning, especially on complex tasks? Without explainability, the work lacks depth, making it difficult to assess the model’s true potential.
>
> While a mechanistic understanding of how the model makes its decisions is beyond the scope of this article, we’re glad you find these questions interesting and we hope that our paper might inspire other researchers to start investigating these types of questions across different video models.
>
> > 5\. The compelling argument for video models as general-purpose vision foundations is critically undermined by the *lack of discussion on practical viability*. The claim of potentially replacing efficient, specialized models hinges not only on capability but also on computational efficiency. We strongly recommend the authors address this by including a discussion on the inference cost and latency observed in this study (e.g., via API pricing and generation times), and to explore pathways for future efficiency improvements (e.g., via model distillation or quantization). Without this, the "replacement" thesis lacks credibility and practical guidance for the community.
>
> We completely agree that video models are not viable substitutions today. **We now include API pricing and generation times in the newly added App. D** and have **adapted the discussion section** (paragraph “Video generation is expensive, but costs tend to fall”) to better reflect the current practicability of video models. Does this address your comment?

---

### Official Review · Reviewer_F9Hr · 2025-10-30

**Soundness:** 2
**Presentation:** 4
**Contribution:** 3
**Rating:** 6
**Confidence:** 4

**Summary:**

This paper explores whether modern video generation models, like Veo3, have zero-shot generalization and reasoning capabilities akin to those of LLMs for langauges. After evaluating on 62 qualitative and 7 quantitative tasks, the authors argue that video models can solve a diverse range of vision tasks, including perception, modeling, manipulation, and reasoning, in zero-shot. Therefore, the paper positions Veo 3 as a potential “foundation model” for vision, analogous to how GPT3 in NLP. The paper is extremely well written, engaging, and comprehensive.

**Strengths:**

- Clarity and structure: The paper is exceptionally clear and easy to follow. The hierarchy of abilities (perception → modeling → manipulation → reasoning) is intuitive and pedagogically effective.
- Breadth of evaluation: The authors test Veo 3 on an impressively wide range of tasks, demonstrating both creativity and thoroughness.
- Novel perspective: The analogy comparing the NLP trajectory from task-specific models to LLMs is both thought-provoking and pertinent to the vision community.

**Weaknesses:**

- Lack of standardized automatic metrics: While 7 quantitative tasks are included, most evaluations rely on human eyes instead of automatic metrics. This is problematic for claims of “zero-shot reasoning.” More automatic and reproducible metrics (PNSR, SSIM, LPIPS, etc.) for different downstream tasks respectively would strengthen the argument.
- Limited style and modality diversity: Most examples appear to use realistic-style image. It is unclear whether Veo 3’s performance generalizes to other styles (cartoon, sketch, abstract, etc.) or to non-realistic renderings. Since the claim is about general vision reasoning, this stylistic limitation weakens the universality of the conclusion.
- Overgeneralized framing: The title and abstract make sweeping claims (“video models are zero-shot reasoners”) that exceed the evidence presented. While current experiments demonstrate some results, they lack solid metrics to substantiate these claims.

**Questions:**

- Could you please provide the total number of unique images? There is a statistical analysis in Appendix B, but Figures 10, 11, and so on show the same image for various downstream tasks. Will this introduce biases into the model, such as making it more susceptible to performing tasks with this specific image?

---

> ### Author Response · Authors · 2025-11-20
> **Response to reviewer F9Hr (1/2)**
>
> Dear reviewer F9Hr,
>
> Thank you for your insightful comments, and for praising the **novel perspective** and **breadth of evaluation** our work constitutes. We're also pleased to hear that you found the manuscript **exceptionally clear and easy to follow**, and hope that the additions based on your feedback further strengthen our contribution.
>
> > Lack of standardized automatic metrics: While 7 quantitative tasks are included, *most evaluations rely on human eyes instead of automatic metrics\. This is problematic for claims of “zero-shot reasoning.” More automatic and reproducible metrics (PNSR, SSIM, LPIPS, etc.) for different downstream tasks respectively would strengthen the argument.
>
> This might be a misunderstanding. While our *qualitative* analysis relies on human judgement, our *quantitative* analysis does not.
>
> Most tasks in the qualitative section can unfortunately not readily be assessed with existing metrics. There are three reasons:
>
> 1. Metrics like PNSR etc. are image metrics. Many of our qualitative tasks require not only the evaluation of a single frame, but of the entire video.
> 2. PNSR etc. are computed relative to a ground truth. For most tasks, there is no single ground truth as tasks can be solved in a variety of ways.
> 3. Pixel-level metrics are insufficient to evaluate most tasks. E.g., solving a maze slowly or quickly, or backtracking along the way, would lead to different videos.
>
> These issues are common in evaluating generated videos. For this reason, we used human raters where applicable and also developed or used high-quality, automated, task-specific metrics for six different *quantitative* tasks that don’t require human intervention or raters. Those tasks span 17,580 videos for a total of 140,640 seconds or 39 hours of generated video. An analysis of video models at this scale is, to the best of our knowledge, extremely rare, and we were happy to see that all four reviewers (including yourself) highlighted the “comprehensive empirical study” (nXQ4) / “Breadth of evaluation” (F9Hr) / “exceptionally high quality of the empirical evidence” (k3zu).
>
> > Limited style and modality diversity: Most examples appear to use realistic-style image. It is *unclear whether Veo 3’s performance generalizes to other styles (cartoon, sketch, abstract, etc.) or to non-realistic renderings*. Since the claim is about general vision reasoning, this stylistic limitation weakens the universality of the conclusion.
>
> We agree that it’s important to show generalization across a rich variety of styles. For this reason, our submission includes a comprehensive analysis in the appendix: 13 of our qualitative tasks (Rorschach blots, Omniglot tasks, Graph traversal, Tree BFS, Sequence completion, Connecting colors, Sorting numbers, Sudoku, Water puzzle, Maze solving, Rule extrapolation) and 3 of our quantitative tasks (Maze Solving, Symmetry, Analogy) are entirely abstract, featuring simple geometric shapes or symbols.
>
> 16 additional tasks (Edge detection, Segmentation, Keypoint localization, Super-resolution, Deblurring, Denoising, Dalmatian illusion, Shape cue-conflict, Background removal, Style transfer, Colorization, Inpainting, Text manipulation, Doodle editing, Scene composition, Transfiguration) feature a combination of natural images and abstract shapes, text, or, cartoon/sketches. Together, they provide evidence that Veo’s capabilities extend beyond real-world images. If you have further suggestions for tasks / styles we should test, kindly let us know.

---

> ### Author Response · Authors · 2025-11-20
> **Response to reviewer F9Hr (2/2)**
>
> > Overgeneralized framing: The title and abstract make sweeping *claims (“video models are zero-shot reasoners”) that exceed the evidence presented*. While current experiments demonstrate some results, they lack solid metrics to substantiate these claims.
>
> We’re not quite sure we fully understood your concern, but we welcome your feedback. For instance, solving a maze requires visual reasoning / planning, and our article analyzes 8,000 maze solution videos across four different sizes and splits (regular vs. irregular, 5x5, 7x7, 9x9). In order to assess performance, we developed an automated and fully deterministic maze-solving metric (akin to a verifier, explained in the corresponding appendix section) that has perfect accuracy by design, since only valid solutions subject to strict criteria are scored as a success. This is just one example that highlights how statements in the paper (in this case: zero-shot maze solving, a form of visual reasoning) are backed up by comprehensive experiments and solid, automated metrics. We recognize that many details on our metric development are presented in the appendix as opposed to the main paper—would it be helpful if we dedicate more space in the main paper to explain our automated metrics?
>
> > Could you please provide the *total number of unique images*? There is a statistical analysis in Appendix B, but Figures 10, 11, and so on show *the same image for various downstream tasks*. Will this introduce biases into the model, such as making it more susceptible to performing tasks with this specific image?
>
> **We've now included a breakdown in Table 1** (Appendix B). There are a total of 930 unique images used: 884 across quantitative tasks, plus 46 across the qualitative tasks. To clarify, since the model isn’t trained or adapted for any of the test images, it is impossible for the test images to influence or bias the model. Reusing the same starting frame for some tasks was done to avoid the impression of a “cherry-picked” starting frame that is different for each task.

---

> > ### Comment · Reviewer_F9Hr · 2025-11-26
> >
> > Thank you for your detailed rebuttal. However, I respectfully note that two of my core concerns remain unaddressed.
> >
> > 1. Although you mentioned the challenges of using auto metrics in all downstream tasks, for some tasks like segmentation (already shown in the paper), edge detection, and so on, it is possible to do the evaluation at the final frame or the successful frame.
> > 2. I agree that 17,580 videos are very promising. However, they start from only 930 unique images. Will the annotators see the same image for different tasks? Does this introduce bias to the evaluation?
> >
> > Therefore, while I appreciate the effort, the fundamental weaknesses persist, and my original positive assessment remains appropriate.

---

> > > ### Author Response · Authors · 2025-11-26
> > >
> > > Dear Reviewer F9Hr,
> > >
> > > Thanks for taking the time to reply!
> > >
> > > 1. Indeed, it is possible to evaluate generated videos at the final frame or across different frames. This is a general property of evaluating videos, not a drawback of our approach; we report both to provide a comprehensive assessment of a model’s abilities.
> > >
> > > 2. Annotators rate generated videos against clearly defined success criteria (now detailed in Appendix A.5). While some of these videos have the same starting frame, all videos are unique and generated from a different random seed. You’re raising an important question: If an annotator sees the same starting frame for two different tasks (e.g., first for edge detection and then for keypoint detection), does this introduce bias to the evaluation? In other words, would the annotator rate the second task (e.g., keypoint detection) differently if they hadn’t seen the same starting frame already for edge detection? If there is such a bias, this would mean that raters who see tasks in a different order would rate these tasks differently; and consequently, inter-rater agreement would be low. Fortunately, our human data shows that this is not the case: inter-rater agreement as measured by Cohen’s kappa is extremely high (k > 0.8, “strong to almost perfect agreement”), thus there is qualitative evidence that this does not introduce bias to the human evaluation. Since other readers might have the same question, we'd be happy to describe this in our article, would that be helpful?
> > >
> > > Does this address your concerns?

---

### Official Review · Reviewer_nXQ4 · 2025-10-31

**Soundness:** 2
**Presentation:** 3
**Contribution:** 2
**Rating:** 2
**Confidence:** 4

**Summary:**

This paper argues that large-scale generative video models are on a trajectory to become general-purpose foundation models for machine vision, much as LLMs did for NLP. The authors' core claim is that these models, trained on simple generative primitives at scale, exhibit emergent zero-shot capabilities for tasks they were not explicitly trained on. The methodology consists of prompting the publicly available Veo 3 model  (and its predecessor, Veo 2) with an initial image and a text instruction. The authors then analyze the model's performance on a massive qualitative benchmark of 62 tasks and a focused quantitative benchmark of 7 tasks. The primary finding is that Veo 3 demonstrates broad zero-shot abilities across this stack, substantially outperforming Veo 2 and proving competitive with strong, specialized image models (like Nano Banana) on several tasks. The authors propose the "Chain-of-Frames (CoF)" concept, analogous to Chain-of-Thought, to explain how frame-by-frame generation enables spatio-temporal reasoning.

**Strengths:**

- This paper is well-written and easy to follow.
- Comprehensive evaluation of 62 tasks covering multiple aspects.
- Showing strong results, in a zero-shot manner, and covering multiple aspects.

**Weaknesses:**

The major weakness is that the analysis is specifically conducted under the Veo model, which has the following limitations:
- Due to a lack of training and designing details, it is impossible to disentangle the single (or even multiple) factors that enable the strong performance of veo model. In other words, it is not surprising that large-scale video generation models show strong results for zero-shot video generation; a few studies also show that these models yield superior performance on non-generation tasks.
- It's hard to verify whether the conclusion is applicable to Veo only or any SOTA video generation models (e.g., SORA)

Some minor weaknesses:
- The success rate for the 62 qualitative tasks was "determined by the authors", which is highly subjective;
- Also, the claim of zero-shot is also suspicious. Without knowing the training data of veo model, it's hard to detect whether some of the tasks are actually "zero-shot".

**Questions:**

- The prompt rewriter is the most significant confound. What would the results be if the prompt rewriter is turned off?


---
Justification:
This paper presents a fascinating and comprehensive empirical study of the emergent zero-shot capabilities of the Veo 3 video. The breadth of the 62-task qualitative analysis is impressive, and the "Chain-of-Frames" concept is a valuable contribution to how we think about visual reasoning. The clear demonstration of scaling-driven improvement from Veo 2 to Veo 3 supports the central thesis.

However, the paper suffers from two major flaws that make it a poor fit for ICLR:
- it is fundamentally a technical report analyzing a closed-source commercial product; it offers no algorithmic novelty or reproducible research contribution.
- Its methodology is confounded by the use of a black-box LLM prompt rewriter, making it impossible to attribute the "reasoning" abilities solely to the video model. The "zero-shot" claims are also weakened by the admission of extensive, expert-level prompt engineering.

---

> ### Author Response · Authors · 2025-11-20
> **Response to reviewer nXQ4: added more experiments, and discussion in the manuscript (1/3)**
>
> Dear reviewer nXQ4,
> Thank you for your time and feedback. We hear your concerns, appreciate your feedback and were happy to read that you found our evaluation **massive, fascinating, and comprehensive**, with **strong results** that **clearly demonstrate scaling-driven improvements** and **support the central thesis**.
>
> Based on your insightful comments, we have run additional experiments and added multiple sections to the manuscript, which we detail below.
>
> > It's hard to verify whether the _conclusion is applicable to Veo only or any SOTA video generation models_ (e.g., SORA):
>
> We share your interest in understanding whether the strong performance of Veo 3 across tasks it wasn’t trained or adapted for is limited to Veo, or whether other strong video generation models show similar patterns. While Sora 2 was only released after the ICLR abstract deadline, a more recent article inspired by our work (\[1\], released a few days ago) has investigated similar visual reasoning tasks in Sora 2, finding strong results. Furthermore, another recent article (\[2\], also released after the ICLR deadline) finds that an open-source model, CogVideoX1.5-5B, performs well on visual reasoning tasks when combined with a lightweight adapter; and \[3\] released a couple of days ago - directly building on and inspired by our work - shows evidence of visual reasoning, to varying degrees, in Sora-2, Hailuo-02, Veo-3.1, Kling-2.5-Turbo-Pro, Vidu-Q2, and Seedance-1.0-Lite. Together, these three articles show that the strong performance we find isn’t limited to a single model family. In light of your feedback, **we’ve now updated our article to include a discussion on these related and important findings** across different video model families (paragraph “Will other video models show similar behavior?” in Section 5).
>
> \[1\] [https://arxiv.org/abs/2511.04570](https://arxiv.org/abs/2511.04570)
> \[2\] [https://arxiv.org/abs/2510.24448](https://arxiv.org/abs/2510.24448)
> \[3\] [https://arxiv.org/pdf/2511.16668](https://arxiv.org/pdf/2511.16668)
>
> > Due to a lack of training and designing details, it is *impossible to disentangle the single (or even multiple) factors that enable the strong performance* of veo model.
>
> We fully agree that it is valuable to disentangle the factors that contribute to strong performance. Factors that might play a role include the data mixture, the model architecture, the training recipe (including pretraining and post-training), the prompt rewriter (discussed in more detail below), and the verbal and visual prompts. While our article is focused on a single model family (Veo), as described above, three different model families developed by different organizations / communities (Veo: Google DeepMind, Sora 2: OpenAI, CogVideoX: open-source community) all demonstrate very promising results on e.g. visual reasoning tasks. This indicates that there doesn't seem to be a “secret sauce”: Some degree of performance can be expected independent of architectural details, specific training recipes, or the exact data mixture(s) used. Instead, this type of behavior simply seems to emerge from training large video models on large video datasets. **We’ve now included this discussion in Sec. 5** (paragraph “Will other video models show similar behavior?”). Additionally, **we have added a discussion in Sec. 5** (paragraph “Where does Veo's performance come from”), which we explain in our response to reviewer k3zu.
>
> > It is *not surprising* that large-scale video generation models show strong results for zero-shot video generation; *a few studies also show* that these models yield superior performance on non-generation tasks.
>
> Would you mind providing references? For instance, are you aware of any articles that demonstrate zero-shot segmentation, zero-shot edge detection, zero-shot maze or symmetry solving with a video model? To the best of our knowledge, the sheer breadth and complexity of tasks that can be solved in a zero-shot manner by a generative video model have not been demonstrated before; e.g., other reviewers described our findings as “remarkable zero-shot capabilities”, “a novel conceptual contribution to the field” (k3zu), and a “novel perspective” (F9Hr).
>
> > It is fundamentally a *technical report analyzing a closed-source commercial product*.
>
> Should the reviewer be aware of an ICLR policy that prohibits analyzing closed-source frontier models, we would welcome a pointer. Since ICLR, NeurIPS and ICML regularly publish articles that analyze frontier models, including commercial ones like Claude, GPT-4, or Gemini, there clearly seems to be substantial scientific interest in understanding these models, their capabilities and limitations, across different tasks. Other reviewers agree—e.g., reviewer qvjV wrote “investigating their \[video foundation model\] abilities and shortcomings is of great interest to the scientific community”.

---

> ### Author Response · Authors · 2025-11-20
> **Response to reviewer nXQ4: added more experiments, and discussion in the manuscript (2/3)**
>
> > It offers *no algorithmic novelty or reproducible research contribution*.
>
> Since our paper is an analysis contribution, we don’t claim algorithmic novelty (just like other analysis articles published at ICLR/NeurIPS/ICML). The core research contribution of our article is showing that a capable video model can solve a broad variety of visual tasks, without task-specific training or adaptation, suggesting a path towards video models as visual foundation models. There is significant community interest in this finding: Following the arXiv release of this article a few months ago, it has already seen several follow-up scientific articles (and even seen interest from the wider community such as an article in The Economist and 250,000 views of a YouTube summary about the article \- of course our paper needs to be assessed as a standalone scientific contribution, but this goes to show that there is significant community interest in our analysis findings). Based on extensive experiments and analyses, our article advocates for a new research direction: advancing visual intelligence through video models. As such, we are particularly keen to see how the research community will explore related questions and build on our analyses. Other reviewers believe that our work provides a **novel perspective** (F9Hr) with **remarkable originality** (k3zu) that is **thought-provoking**, **pertinent**, and **of great interest** to the community (F9Hr, qvjV).
>
> In terms of reproducibility, is your concern that the model we investigate isn’t open-source? If so, our research contribution isn’t the model itself (we don’t propose or train the model), but an analysis of one of the world’s best video models. The analysis itself is indeed a reproducible research contribution: We used a publicly available API (Vertex) for our analyses similar to how thousands of ICLR/NeurIPS/ICML articles query the Claude/GPT/Gemini APIs. Anyone can run the experiments that we ran, using the prompts that we report in our article, and replicate the results.
>
> > The success rate for the 62 qualitative tasks was "determined by the authors", which is *highly subjective*.
>
> The qualitative tasks are described as “a rough estimate” in the caption, “a ballpark estimate” in the text, and as a “qualitative” section overall. They’re intended to provide an idea of the breadth of tasks that can be tackled with a video model, not to be a replacement for the objective quantification in Sec. 4\.
> Nevertheless, we agree that the evaluation could have been more transparent. We **now provide additional annotations**, including from an external computer vision expert, and report the inter-rater reliability (Cohen’s kappa). Concretely, we ask two authors and one external expert to annotate 10 additional samples for each qualitative task (1860 total annotations). The pair-wise rater reliability is 0.82 between authors and 0.87 and 0.89 between authors and external annotator. This is generally considered *strong* to *almost perfect* agreement \[3\], indicating a stable, objective evaluation. We have added this analysis alongside the exact guidelines used to determine success in each task in the **newly added App. A.5**.
> \[3\] [https://pmc.ncbi.nlm.nih.gov/articles/PMC3900052/](https://pmc.ncbi.nlm.nih.gov/articles/PMC3900052/)

---

> ### Author Response · Authors · 2025-11-20
> **Response to reviewer nXQ4: added more experiments, and discussion in the manuscript (3/3)**
>
> > The claim of zero-shot is also suspicious. Without knowing the training data of veo model, it's *hard to detect whether some of the tasks are actually "zero-shot"*.
>
> Thanks for this question\! We previously hadn’t explained how the term “zero-shot” is defined in our article.
> TL/DR: We follow the contemporary definition that’s used in NLP in the context of LLMs.
>
> Longer explanation, **now added as a standalone paragraph in the discussion section** in response to your question:
>
> “Zero-shot learning in the era of web-scale data: The advent of models trained on web-scale data has led to a necessary evolution in the understanding of zero-shot capabilities. Given the vast and diverse nature of the training data, it is difficult to verify that a model has not encountered data similar to a particular task before. In contemporary usage, the definition of “zero-shot learning” has therefore shifted from a strict “unseen task” criterion (often impossible to verify) to a methodological one \[1, 2, 3\] which we follow here: A task is considered zero-shot if the model can be prompted to perform the task despite not being trained, adapted, or fine-tuned specifically for that task. For instance, LLMs were described as “zero-shot reasoners” \[1\]  not because of any guarantees that reasoning-style text isn't part of the training data by accident, but because they can perform reasoning tasks without task-specific training or adaptation, simply via prompting alone. The ability of Veo 3 to solve a wide range of tasks in this manner indicates a paradigm shift in machine vision, mirroring the one that occurred in natural language processing.”
>
> \[1\]Large language models are zero-shot reasoners
> \[2\]Sparks of Artificial General Intelligence: Early experiments with GPT-4
> \[3\] Pre-train, Prompt, and Predict: A Systematic Survey of Prompting Methods in Natural Language Processing
>
> > The *"zero-shot" claims are also weakened by the admission of extensive, expert-level prompt engineering*.
>
> With regards to prompt engineering: thank you for your thoughtful question. We agree that the role of prompt engineering merits a clear explanation. Our use of prompt engineering is a standard and necessary practice for evaluating the capabilities of large foundation models. Since these models are not fine-tuned for specific tasks, a textual prompt is the primary mechanism to provide them with instructions. Without good prompts, we would risk underestimating and therefore misrepresenting a model’s abilities. Similarly, with frontier models, prompt engineering is a necessary component that enables us to communicate with them in a way that doesn’t misrepresent their abilities. For this reason, we’re very transparent about the prompts we used and also provide a section on best practices for prompts in the Appendix.
>
> > The *prompt rewriter is the most significant confound*. What would the results be if the prompt rewriter is turned off? / Its methodology is confounded by the use of a black-box LLM prompt rewriter, making it impossible to attribute the "reasoning" abilities solely to the video model.
>
> If performance of an LLM alone (no video model involved) is worse than that of an LLM+video model, this leaves two options: either the video model itself is outperforming the LLM, or there is a form of *symbiotic reasoning* between an LLM and a video model. We fully agree that it would be interesting to disentangle these hypotheses. Either way, if the standalone LLM is outperformed, we can conclude that video models are *necessary* components for the visual reasoning performance we report.
>
> Fig. 7 has evidence to understand this better: We compare Gemini's performance given the initial frame (grey dashed line) to that of Veo. Gemini (LLM only) is consistently outperformed by Veo (LLM+video model).
>
> Prompt enhancement/rewriting is a common component of generative video models (e.g., \[4\]). While fully evaluating its exact role is out of the scope of this work, our results clearly show that video models are a necessary component for the strong maze solving results we find. While we have no evidence to suggest that they’re also sufficient components, we hope to inspire others to test video-only models on these tasks as well, which could shed further light on these questions.
>
> \[4\] [https://github.com/Lightricks/LTX-Video](https://github.com/Lightricks/LTX-Video)

---

> ### Comment · Reviewer_nXQ4 · 2025-11-26
> **Response to authors**
>
> I thank the authors for their detailed response and the significant effort put into running additional experiments, particularly the inclusion of external expert annotators to calculate inter-rater reliability for the qualitative tasks. This addresses my concern regarding the subjectivity of the initial evaluation. I also appreciate the inclusion of discussions regarding concurrent works on other video models.
>
> However, my primary concerns regarding the methodology and the specific claims of "reasoning" remain unresolved:
>
> 1. The Prompt Rewriter Confound The authors acknowledge that the prompt rewriter cannot be disabled. The rebuttal argues that because Veo (Rewriter + Video) outperforms Gemini (LLM), the video model is a "necessary component." While true, "necessary" does not mean the video model is performing the reasoning. If the hidden prompt rewriter (an LLM) solves the logic of the task (e.g., solving the maze textually or defining the symmetry) and passes a highly detailed visual description to the video model, the video model acts primarily as a renderer, not a reasoner. Without an ablation study turning off the rewriter, attributing "Zero-Shot Reasoning" capabilities specifically to the video model is scientifically unsound.
>
> 2. The "Black Box" Limitation While the authors argue that analyzing frontier models is of community interest, the inability to inspect the training data or architecture limits the scientific insights to observations rather than explanations. The claim of "Zero-Shot" relies on a definition ("not explicitly trained for") that is impossible to verify without access to the training data mixture. It remains indistinguishable whether the model is reasoning or simply recalling similar visual puzzle distributions from web-scale training data.

---

> ### Author Response · Authors · 2025-11-28
>
> Dear reviewer nXQ4,
>
> Thanks for following up\! We’re glad to hear that our response addressed some of your major concerns from the initial review.
>
>
> >*“... does not mean the video model is performing the reasoning. If the hidden prompt rewriter (an LLM) solves the logic of the task (e.g., solving the maze textually or defining the symmetry) and passes a highly detailed visual description to the video model, the video model acts primarily as a renderer, not a reasoner.”*
>
> Thanks for raising this, we fully agree\! If an LLM can solve the task, then the video model could simply “render” the LLM-provided solution visually. However, the reverse is also true: if an LLM cannot solve the task, and a video model (with LLM prompt rewriter) can solve the task, then the solution must come from the video model, not the prompt rewriter. This is exactly what we find in Figure 7: Veo 3 achieves a pass@10 of 78% for 5x5 mazes, 38% for 7x7 mazes, 12% for 9x9 mazes, and 75% for irregular mazes whereas Gemini 2.5 Pro with image input achieves 16%, 8%, 0%, and 0% respectively on the same mazes. Further, **in response to your concern, we also just tested Gemini 3.0 Pro (currently the world’s best LLM according to LMArena) on a random set of 9x9 mazes, finding that the language model still performs at chance (0%). Hence, the solution here must come from the video model itself, not from a language model.** Additionally, dense perceptual tasks like edge detection or segmentation cannot be solved with a language model either (which we’re happy to provide evidence for if you think it would be helpful).
>
>
> >*“the inability to inspect the training data or architecture limits the scientific insights to observations rather than explanations”:*
>
> We would like to clarify that reporting & analyzing empirical observations is precisely the goal of our submission. Science often starts from reporting intriguing phenomena (observations), which later turn into hypotheses/explanations, which later turn into a theory. Each of these steps is important; our observations in particular are being regarded as highly insightful by the community. E.g., without being involved in this work, an Antrophic co-founder blogged “The implications \[of our article\] are profound – I expect we’re going to get extremely smart, capable robot ‘agents’ through the development of smart and eventually distilled video models.”; our analyses were featured by The Economist & in a YouTube with 250,000 views. Furthermore, our analyses already inspired several follow-up articles that investigate similar phenomena across Sora-2, CogVideoX1.5-5B, Hailuo-02, Veo-3.1, Kling-2.5-Turbo-Pro, Vidu-Q2, and Seedance-1.0-Lite. All of these have different architectures and training data; **the fact that similar observations hold across models (as we now discuss in Section 5\) provides** **strong evidence that architecture / training data do not play a decisive role**.
>
> >*“The claim of "Zero-Shot" relies on a definition ("not explicitly trained for") that is impossible to verify without access to the training data mixture. It remains indistinguishable whether the model is reasoning or simply recalling similar visual puzzle distributions from web-scale training data.”:*
>
> We understand the reviewer's concern regarding the "black box" nature of the model and its implications for our zero-shot claims. As researchers, our analysis is necessarily based on the information made publicly available by the model's developers. In the absence of specifics about the training data, we must ground our analyses in the observable behaviour.
>
> Given that we constructed our visual reasoning test cases (e.g. maze / symmetry solving) months after Veo 3 was released, **we can confirm with certainty that it is impossible for these exact test cases to have been part of the training mix**. Recalling similar puzzles would not help since we randomized the maze layout and start/ target positions and we also randomized our symmetry task patterns. Some of the model developers have, in fact, contacted us after seeing our paper, mentioning how surprised they were to see these \- quote \- “impressive results” across perception, modeling, manipulation and reasoning from a model developed to generate entertaining videos. In light of this, **our work aligns with both the original and the contemporary definition of zero-shot learning** \[1, 2\]: the original definition is the ability to perform a task without having been explicitly trained on labeled examples for that specific task; and the contemporary definition \[e.g. 3, 4\] is used for large-scale models that can be prompted to perform a task with zero examples in the prompt, without task-specific fine-tuning or adaptation.
>
> \[1\] Importance of Semantic Representation: Dataless Classification (2008)
> \[2\] Zero-Shot Learning with Semantic Output Codes (2009)
> \[3\] Language models are zero-shot reasoners
> \[4\] On the opportunities and risks of foundation models

---

### Author Response · Authors · 2025-11-20
**General response to reviewers**

We thank all reviewers for their insightful comments. We very much appreciate the consensus that our study covers a **highly relevant and important topic** (qvjV) from a **novel perspective** (F9Hr) with **remarkable originality** (k3zu), **creativity and thoroughness** (F9Hr). As such, it **possesses substantial significance**, providing **compelling evidence for an impending paradigm shift in computer vision** (k3zu) and is **thought-provoking**, **pertinent**, and **of great interest** to the community (F9Hr, qvjV).

Reviewers note the **massive** (nXQ4, k3zu) and **extensive** (k3zu, qvjV) benchmark with an **impressively wide range of tasks** (F9Hr) and the additional **valuable contribution** of the chain-of-frames concept (nXQ4). The reviewers agree that our experiments serve as a **clear demonstration of scaling-driven improvement** (nXQ4) and that the **core argument is convincing and well-supported** (k3zu, nXQ4).

That said, reviewers also identified several important opportunities for improvement, addressed as follows (more details and updates in individual replies):

- Qualitative tasks: we now provide evidence that the annotation criteria (now clearly set out in App. A.5) lead to consistent, non-subjective ratings based on 1860 additional human annotations and very high inter-rater reliability.
- Expanded comparison to image generation model (Nano Banana) now encompassing all visual reasoning tasks.
- Substantially expanded and revised discussion (Section 5\) with four new paragraphs: (1.) “Zero-shot learning in the era of web-scale data” defining our use of the term, (2.) “Will other video models show similar behavior?” explaining how the strong results we find aren’t limited to Veo, (3.) “Where does Veo's performance come from?” formulating our hypotheses and outlines directions for future research, and (4.) “Similarities and differences with NLP” with a discussion on parallels and differences between the development of visual vs. language foundation models.

We thank all reviewers for their constructive feedback; updates are highlighted in the manuscript with a light blue color to make them easy to find.

---

### Meta-Review · Area_Chair_KsqQ · 2026-01-07

**Summary:**

This is a very intriguing work. If the claims were true, then it could well be a very seminal paper among a body of several others (as mentioned by the authors). Some of the reviewer concerns were potentially major/fatal issues, particularly the definition or evidence of being "zero-shot". The authors admit that it can't be proved the Veo3 wasn't trained on their tasks, but argue that the well-known paper about LLMs as being zero-shot reasoners took a functional approach/definition. I can see some logic in their argument, but am only partially convinced.

What is more germaine to me is that the actual results are only sometimes/partially convincing. If the authors go with the functional definition, then their argument hinges on really strong results. However, the K=1 results are only sometimes strong, and looking at up to K=10 results is hardly convincing of Veo3 being a strong "zero-shot reasoner". Furthermore, certain results are fairly poor even up to K=10, e.g. maze solving, visual symmetry (last frame), visual analogy (reflect and rotate are below chance!), etc.. The authors argue that the performance bump from Veo2 to Veo3 is significant, just like how LLMs improved rapidly since GPT-3 (and so now nobody doubts the zero-shot claim for LLMs since ChatGPT onwards). I would counter-argue that one cannot accept scientific claims based on projected/assumed future results.

Also pertinently, the entire paper is based on Veo3's results -- one model. I agree with one reviewer that this makes the current work a technical report more than a scientific paper. The authors argue that since submission, a few other models have shown similar properties. The claim of the paper (in its title) is about video models (**plural**); when multiple models are run on this benchmark, and show strong results, then such a claim is better supported.

That said, I would perhaps have taken a risk and recommended acceptance if the work had some component of mechanistic explaininability or even some rigorous analysis towards scientific understanding. Perhaps even if the work were framed as a dataset/benchmark plus a suggestive claim, rather than a qualified claim ("video models are on a path to becoming") that is not really supported.

Utimately, I cannot recommend acceptance for the work in its current form as a scientific submission to a highly-selective peer-reviewed conference. That said, I have little doubt that the research community would explode with excitement/discussion if/when this work is released as an arxiv paper or social media post, etc. Perhaps that is the best venue for the current form and maturity of this work.

**Reviewer Concerns:**

Pls see above.

**Reviewer Scores:**

-- F9HR: explicitly maintained score of 6.
-- nXQ4: implied that score remains at 2.
-- qvJV and k3zu: 50/50 chance raised the scores.

---

### Decision · Program_Chairs · 2026-01-26

Reject